# LiRDSC: Ligand-Conditioned RNA Sequence Design via Diffusive Structural Conditioning

## Abstract

Designing RNA sequences that bind specific small-molecule ligands is a central goal in molecular engineering. However, existing computational methods face two persistent challenges: extreme sensitivity to imperfections in input tertiary structure scaffolds, and a tendency toward *mode collapse*, where models generate generic, non-specific sequences rather than ligand-tailored designs. To address this, we present two core contributions. First, we introduce **RLData2400**, a benchmark dataset combining high-resolution experimental structures with diverse, high-confidence *in silico* models to facilitate the development of models. Second, we propose **LiRDSC (Ligand-conditioned RNA Design via Diffusive Structural Conditioning)**, a deep generative framework architected for specificity. LiRDSC uniquely employs a **Diffusive Structural Encoder (DSE)**, which learns resilient representations by training on noise-perturbed structures, and a **Ligand-Contextual FiLM Conditioner (LCFC)** that steers the model to reason about the ligand's identity, preventing mode collapse. Trained on our dataset (RLData2400), LiRDSC not only achieves high sequence recovery but also generates a diverse range of ligand-target RNA sequences. Crucially, its superiority is most pronounced on structurally augmented data, directly validating the robustness imparted by our diffusion-based conditioning. Inverse folding experiments further confirm that the generated sequences accurately recapitulate their target tertiary structures. Importantly, computational analysis predicts strong binding compatibility with the intended ligands, demonstrating LiRDSC's ability to produce RNA candidates that are both structurally viable and ligand-specific.

## 1 Introduction

The ability to design functional RNA molecules that bind to specific small-molecule ligands is a cornerstone of molecular engineering, with profound implications for creating synthetic riboswitches, therapeutic aptamers, and novel biosensors (Falese et al., 2021; Kovachka et al., 2024; Wong et al., 2024). The central challenge in this field is the *inverse design problem*: computationally generating an RNA sequence that not only folds into a predetermined three-dimensional structure but also forms a stable, functional complex with a specific small-molecule ligand (Tan et al., 2023; Huang et al., 2024). While classical combinatorial and physics-based methods have laid the groundwork (Laganà et al., 2014), the field has rapidly shifted towards data-driven deep learning approaches.

Despite recent advances in deep learning, computational approaches are persistently hindered by two fundamental obstacles (Ching et al., 2018). First, the performance of structure-based models is often extremely sensitive to the fidelity of the input tertiary structure (Singh et al., 2022), making them brittle to the inevitable noise or inaccuracies present in both experimental and predicted structural scaffolds. Second, generative models frequently suffer from "mode collapse", a phenomenon where they tend to produce generic, high-frequency sequences that are structurally plausible but lack the chemical and steric specificity required for binding a particular ligand, which is a known issue in conditioned generation for proteins that remains nascent for RNA (Anand et al., 2022).

To address these critical limitations, we introduce **LiRDSC**, a deep generative framework architected for specificity. LiRDSC confronts the challenge of structural noise head-on by employing a **Diffusive Structural Encoder**. This component learns resilient structural representations by being explicitly trained on noise-perturbed versions of RNA backbones, thereby acquiring robustness

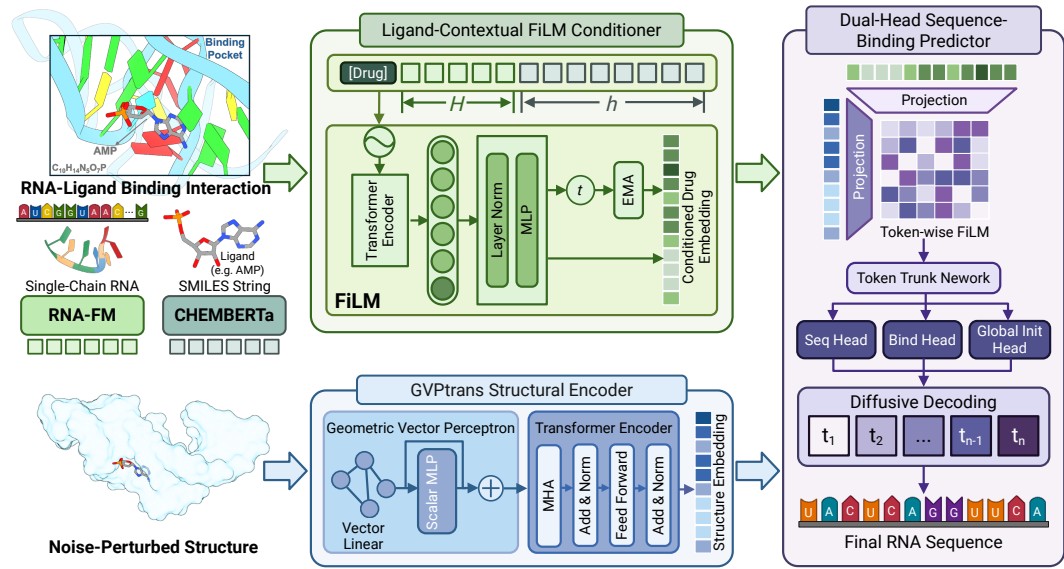

Figure 1: Overview of the **LiRDSC** framework. LiRDSC takes a noise-perturbed RNA structure and a ligand SMILES string as input. A **GVPtrans Structural Encoder** generates a robust structural representation ($H_{\text{struct}}$), while a **Ligand-Contextual FiLM Conditioner (LCFC)** creates a context-aware ligand embedding ($h^*$). These are fused by a **Dual-Head Sequence-Binding Predictor (DSBP)** using a FiLM layer to inform a final **diffusive decoding** step, which generates the ligand-conditioned RNA sequence.

against structural imperfections. To counteract mode collapse and enforce ligand specificity, we propose the **Ligand-Contextual FiLM Conditioner**, a mechanism that dynamically modulates the structural encoding process with information derived from the ligand's chemical identity. This steers the model to reason about the specific ligand at every position, steering the generative process toward ligand-compatible sequences. Foundational to our work is the development of **RLData2400**, a benchmark dataset of 2,400 RNA-ligand complexes, which uniquely combines high-resolution experimental structures with a diverse set of high-confidence *in silico* models to facilitate the training of generalizable models.

We conduct extensive experiments to validate the effectiveness of LiRDSC, comparing it against state-of-the-art (SOTA) baseline models. Our results demonstrate that LiRDSC not only achieves significantly higher sequence recovery rates but also generates a diverse array of novel sequences. Crucially, its performance superiority is most pronounced when evaluated on the structurally augmented portion of our dataset, directly confirming the robustness imparted by the diffusive structural conditioning. Furthermore, inverse folding experiments show that the sequences designed by LiRDSC accurately recapitulate their target tertiary structures, and critically, computational analysis shows that these designed molecules are predicted to form stable complexes with their intended ligands, validating our approach's ability to produce not only structurally viable but also highly specific RNA candidates for target ligands.

The main contributions of this work are summarized as follows:

- We construct and release **RLData2400**, a new, large-scale benchmark dataset for RNA-ligand inverse design, comprising an equal split of 1,200 high-quality experimental structures and 1,200 high-confidence computationally generated models to promote model robustness.

- We propose **LiRDSC**, a deep generative framework that synergistically integrates a diffusion-based structural encoder with a ligand-contextual conditioner to achieve specific RNA sequence design.

- We introduce two key architectural innovations: the **Diffusive Structural Encoder** to learn noise-resilient representations from imperfect structural data, and the **Ligand-Contextual FiLM Conditioner** to prevent mode collapse and ensure the generation of ligand-specific sequences.

- Through comprehensive experiments, we demonstrate that **LiRDSC** establishes a new state of the art, significantly outperforming existing methods in sequence recovery, structural fidelity of designed sequences, and overall generative diversity.

## 2 DATA CONSTRUCTION

To train and evaluate our model, we construct **RLData2400**, a benchmark dataset comprising 2,400 RNA-ligand complexes. The dataset is intentionally balanced, containing an equal number of experimentally-determined structures and high-confidence computationally-generated models. This hybrid composition ensures that our dataset captures both established biological interactions and diverse, novel structural folds, providing a robust foundation for model development.

### 2.1 EMPIRICAL STRUCTURE CURATION FROM THE PROTEIN DATA BANK

The first component of our dataset consists of 1,200 empirical structures curated from the RCSB Protein Data Bank (PDB) (Rose et al., 2016; Burley et al., 2017). To isolate high-quality and relevant examples, we design and implement a rigorous, multi-stage curation pipeline. The process begins with a broad query for all PDB entries containing the keyword "RNA" and proceeded through several automated filtering stages. The specific steps are as follows:

1. **Initial Retrieval:** A programmatic query is performed for all entries with "RNA" in the "Structure Keywords" field. All resulting entries are downloaded in the `PDB` file format.

2. **Content Verification:** Each file is parsed to confirm the presence of RNA chains and non-polymeric ligand records (`HETATM`). Entries lacking either are discarded.

3. **Ion Exclusion:** To focus on drug-like or metabolic molecules, structures where the only ligands are simple biological ions (e.g., $Mg^{2+}$, $K^+$, $Cl^-$) are filtered out.

4. **Binding Interaction Validation:** A complex is validated as a genuine binding pair only if it satisfied the following distance criterion, a standard metric for non-covalent interactions (Ahmad et al., 2004; Si et al., 2011; Philips et al., 2013):

$$\min_{a_r \in A_R, a_l \in A_L} d(a_r, a_l) \leq 3.5\,\text{Å} \tag{1}$$

   where $A_R$ and $A_L$ are the sets of heavy atoms for the RNA and ligand, respectively, and $d(a_r, a_l)$ is the Euclidean distance. This rule is applied on a per-chain basis, ensuring our dataset consists of single-chain RNA complexes.

5. **Finalization:** All water molecules (`HOH` records) are computationally removed from the coordinate files of the validated complexes.

This pipeline yields a final set of 1,200 high-quality, ground-truth RNA-ligand structures.

### 2.2 *In Silico* DATA AUGMENTATION VIA CONFIDENCE-GUIDED STRUCTURAL PREDICTION

To augment our dataset with novel structures beyond the current experimental record, we develop a method we term **Confidence-Guided Structural Prediction (CGSP)**. This workflow leverages the predictive power of the AlphaFold3 (Jumper et al., 2021; Abramson et al., 2024) to generate an additional 1,200 high-quality, synthetic RNA-ligand complexes. The CGSP workflow is executed as follows:

1. **Sequence Generation:** A library of 1,000 RNA sequences is computationally generated, with lengths sampled uniformly from the integer range [50, 200]. Mechanistically, this length range aligns with the tokenization and processing constraints of both the RNA and ligand encoders (i.e., RNA-FM (Chen et al., 2022), ChemBERTa (Chithrananda et al., 2020)). Meanwhile, it also matches the typical length distribution observed in empirical

RNA–ligand complexes. Additionally, this constraint was selected to maximize predictive confidence from AlphaFold3: in our preliminary tests, shorter RNA sequences (i.e., $\leq 200$ nt) consistently yielded higher predicted Local Distance Difference Test (pLDDT) scores, increasing the likelihood of obtaining structurally plausible RNA-ligand complexes suitable for training.

2. **Ligand Pairing:** Each random RNA sequence is paired with a molecule from the AlphaFold server's built-in ligand pool. This pool contains a variety of biologically significant molecules, including nucleotide phosphates (e.g., ATP, GDP), cofactors (e.g., NAD, FAD), and heme groups.

3. **Structure Prediction:** Each RNA-ligand pair is submitted to the AlphaFold server for tertiary structure prediction. To ensure reproducibility, a fixed random seed (**42**) is used, and the server returned five potential binding poses for each input pair, resulting in an initial pool of 5,000 candidate structures.

4. **Confidence-Based Filtering:** The key step of CGSP is a stringent quality filter based on the pLDDT score. We calculate the mean pLDDT for each of the 5,000 poses and select the top 1,200 complexes with the highest confidence scores. This ensures our synthetic data is composed of structurally plausible models.

5. **Format Standardization:** The final 1,200 selected structures, provided in `mmCIF` format, are converted to the `PDB` format to maintain consistency across the entire **RLData2400** dataset.

The resulting 1,200 *in silico* high-confidence structures form the second half of our balanced benchmark dataset.

## 3 METHODS: LiRDSC

We introduce **LiRDSC** (Figure 1), a modular framework for *de novo* RNA sequence generation conditioned on a ligand and an RNA structural context. LiRDSC couples several components: (i) pretrained models (i.e., RNA-FM (Chen et al., 2022), CHEmberta (Chithrananda et al., 2020)) to generate initial embeddings; (ii) a GVPtrans structural encoder that processes a noise-perturbed structural scaffold; (iii) a Ligand-Contextual FiLM Conditioner that produces a conditioned drug embedding (Perez et al., 2018); and (iv) a Dual-Head Sequence-Binding Predictor which informs a final diffusive decoding step to generate the RNA sequence.

### 3.1 PROBLEM FORMULATION

The goal is to design an RNA sequence $\mathcal{R}$ that folds into a specific tertiary structure and binds to a target small-molecule ligand $\mathcal{L}$. The model takes as input the target RNA backbone structure (from a PDB file), the ligand's chemical representation (e.g., a SMILES string), and for training, the corresponding RNA sequence. The model aims to predict nucleotide logits $\mathbf{Z} \in \mathbb{R}^{B \times S \times 4}$, binding propensities $\hat{\mathbf{y}}^{\text{bind}} \in \mathbb{R}^{B \times S}$, and a global interaction logit $\hat{\mathbf{y}}^{\text{int}} \in \mathbb{R}^{B}$.

### 3.2 INPUT REPRESENTATION AND PRETRAINED EMBEDDINGS

To provide the model with rich initial features, the raw inputs are processed by specialized pretrained models. The RNA sequence is fed into RNA-FM to obtain initial **RNA Embeddings** ($\mathbf{H}^{\text{seq}}$), and the ligand's SMILES string is processed by CHEmberta to get **Ligand Embeddings** ($\mathbf{h}^{\text{lig}}$). The tertiary structure coordinates from the PDB file provide the initial structural fields $\mathbf{X}_0$. During training, these structures are perturbed with noise. We adopt a Variance-Preserving (VP) Stochastic Differential Equation (SDE) to corrupt $\mathbf{X}_0$:

$$\alpha_t^2 = \exp\Big( -\int_0^t \beta(\tau)\, d\tau \Big), \qquad \sigma_t^2 = 1 - \alpha_t^2, \qquad \mathbf{Z}_t = \alpha_t \mathbf{X}_0 + \sigma_t \boldsymbol{\epsilon}, \ \ \boldsymbol{\epsilon} \sim \mathcal{N}(0, \mathbf{I}), \quad (2)$$

where $t \sim \mathcal{U}(0, 1)$ is the noise level, and $\beta(\tau)$ is a predefined noise schedule.

### 3.3 LIGAND-CONTEXTUAL FiLM CONDITIONER

The Ligand-Contextual FiLM Conditioner takes the RNA embeddings ($\mathbf{H}^{\text{seq}}$) and ligand embeddings ($\mathbf{h}^{\text{lig}}$) as input to produce a **Conditioned Drug Embedding**. It first projects them into a common latent space.

$$\mathbf{H} = \mathbf{W}_{\text{seq}}\mathbf{H}^{\text{seq}} + \mathbf{b}_{\text{seq}}, \qquad \mathbf{h} = \mathbf{W}_{\text{lig}}\mathbf{h}^{\text{lig}} + \mathbf{b}_{\text{lig}}, \tag{3}$$

where $\mathbf{W}_{\text{seq}} \in \mathbb{R}^{d \times H_{\text{seq}}}$ and $\mathbf{W}_{\text{lig}} \in \mathbb{R}^{d \times H_{\text{lig}}}$ are learnable projection matrices, and $\mathbf{b}_{\text{seq}}, \mathbf{b}_{\text{lig}} \in \mathbb{R}^d$ are bias terms.

To aggregate the sequence context and inform the ligand representation, we prepend a special [DRUG] token (initialized by the projected ligand embedding $\mathbf{h}$) to the sequence tokens. This combined sequence is then processed by a positional Transformer encoder. The output corresponding to the [DRUG] token serves as a context-aware ligand representation.

$$\tilde{\mathbf{T}} = \mathcal{E}\Big(\text{PE}([\mathbf{h}; \mathbf{H}])\Big), \qquad \mathbf{c} = \tilde{\mathbf{T}}_{[:,0,:]}, \tag{4}$$

where $[\cdot;\cdot]$ denotes concatenation along the token dimension, PE is the sinusoidal positional encoding, $\mathcal{E}$ is the Transformer encoder, and $\mathbf{c} \in \mathbb{R}^{B \times d}$ is the updated [DRUG] context vector.

To modulate the original ligand embedding with the aggregated sequence context, LCFC uses the context vector $\mathbf{c}$ to generate parameters for a Feature-wise Linear Modulation (FiLM) layer.

$$[\boldsymbol{\gamma}, \boldsymbol{\beta}] = \text{MLP}_{\text{film}}(\text{LN}(\mathbf{c})), \qquad \mathbf{h}^* = \boldsymbol{\gamma} \odot \mathbf{h} + \boldsymbol{\beta}, \tag{5}$$

where $\odot$ is element-wise multiplication, LN is Layer Normalization, and $\mathbf{h}^* \in \mathbb{R}^{B \times d}$ is the final conditioned drug embedding.

To enable sequence-agnostic inference, we distill population statistics of the conditional modulation by maintaining exponential moving averages (EMAs) of the FiLM parameters ($\boldsymbol{\gamma}, \boldsymbol{\beta}$).

$$\bar{\boldsymbol{\theta}}^{(t)} = \alpha \bar{\boldsymbol{\theta}}^{(t-1)} + (1 - \alpha) \frac{1}{B} \sum_{b=1}^{B} \boldsymbol{\theta}_b^{(t)}, \qquad \boldsymbol{\theta} \in \{\boldsymbol{\gamma}, \boldsymbol{\beta}\}, \ \alpha \in (0, 1), \tag{6}$$

At test time, we use the averaged parameters $\bar{\boldsymbol{\gamma}}$ and $\bar{\boldsymbol{\beta}}$ to compute the conditioned ligand representation $\mathbf{h}^* = \bar{\boldsymbol{\gamma}} \odot \mathbf{h} + \bar{\boldsymbol{\beta}}$ without needing an input RNA sequence $\mathbf{H}^{\text{seq}}$.

### 3.4 GVPTRANS STRUCTURAL ENCODER

The GVPtrans structural encoder encodes the input tertiary structure into a set of per-residue **Structural Embeddings**. Its input is the noise-perturbed backbone coordinates, $\mathbf{Z}_t$, from Eq. equation 2. The encoder is a hybrid architecture composed of Geometric Vector Perceptron (GVP) layers followed by Transformer blocks. The GVP layers process the tertiary structure geometric data in an SE(3)-equivariant manner, generating initial scalar and vector features for each residue. These features are then passed to a standard Transformer encoder to capture long-range dependencies. This process yields the final structural embeddings:

$$\mathbf{H}^{\text{struct}} = \text{GVPtrans}(\mathbf{Z}_t) \in \mathbb{R}^{B \times S \times H_{\text{struct}}}. \tag{7}$$

### 3.5 DUAL-HEAD SEQUENCE–BINDING PREDICTOR

The Dual-Head Sequence–Binding Predictor fuses the structural embeddings $\mathbf{H}^{\text{struct}}$ and the conditioned drug embedding $\mathbf{h}^*$ to make final predictions. To prepare for this fusion, the structural embeddings are projected into a hidden space, and the conditioned ligand vector $\mathbf{h}^*$ is mapped to generate FiLM parameters.

$$\mathbf{U} = \mathbf{W}_{\text{proj}}\mathbf{H}^{\text{struct}} + \mathbf{b}_{\text{proj}}, \qquad [\boldsymbol{\gamma}^{\text{tok}}, \boldsymbol{\beta}^{\text{tok}}] = \text{MLP}_{\text{tok}}(\text{LN}(\mathbf{h}^*)), \tag{8}$$

with $\mathbf{U} \in \mathbb{R}^{B \times S \times D}$ and $\boldsymbol{\gamma}^{\text{tok}}, \boldsymbol{\beta}^{\text{tok}} \in \mathbb{R}^{B \times D}$.

To enable position-dependent conditioning from the global ligand context, we apply token-wise FiLM. The resulting tensor is then processed by a shared token trunk network $\mathcal{T}$ to generate the final per-residue representations.

$$\tilde{\mathbf{U}} = \eta \, \boldsymbol{\gamma}^{\text{tok}} \odot \mathbf{U} + \boldsymbol{\beta}^{\text{tok}}, \qquad \mathbf{V} = \mathcal{T}(\tilde{\mathbf{U}}), \tag{9}$$

where $\eta > 0$ is a small stability factor and $\mathbf{V} \in \mathbb{R}^{B \times S \times D}$.

To generate the final outputs, two separate linear heads are applied to the shared representation $\mathbf{V}$ to predict nucleotide logits and binding-site propensities.

$$\mathbf{Z} = \mathbf{W}_{\text{seq}}\mathbf{V} + \mathbf{b}_{\text{seq}} \in \mathbb{R}^{B \times S \times 4}, \qquad \hat{\mathbf{y}}^{\text{bind}} = \left(\mathbf{W}_{\text{bind}}\mathbf{V} + \mathbf{b}_{\text{bind}}\right)_{:,:,1} \in \mathbb{R}^{B \times S}. \qquad (10)$$

To encourage overall ligand–RNA compatibility, a third head predicts a global interaction score directly from the conditioned ligand representation $\mathbf{h}^*$.

$$\hat{\mathbf{y}}^{\text{int}} = \mathbf{w}_{\text{int}}^{\top} \sigma\left(\mathbf{W}_{\text{int}}\mathbf{h}^* + \mathbf{b}_{\text{int}}\right) \in \mathbb{R}^B, \qquad (11)$$

where $(\mathbf{W}_{\text{seq}}, \mathbf{b}_{\text{seq}})$ and $(\mathbf{W}_{\text{bind}}, \mathbf{b}_{\text{bind}})$ are linear heads, and $\sigma$ is an activation function like GELU or SiLU.

## 3.6 DIFFUSIVE DECODING

At inference time, we employ a diffusive decoding process, which completely imitates the methodology of RiboDiffusion (Huang et al., 2024), to generate the final RNA sequence. This process iteratively refines a random sequence into a coherent one by treating generation as the reverse of a diffusion process. Starting from a sequence of Gaussian noise, we solve the reverse-time ordinary differential equation (ODE) using the model's predictions as the score function. At each step, the DSBP model is called to predict the denoised sequence, and its output logits $\mathbf{Z}$ guide the update, allowing the generation to be conditioned on both the structural scaffold and the specific ligand context.

## 3.7 LEARNING OBJECTIVE

To train the model, we define a composite loss function that combines objectives for sequence prediction, binding site prediction, and global interaction prediction. For each batch, we sample a noise level $t \sim \mathcal{U}(0, 1)$, compute $(\alpha_t, \sigma_t)$ via the VP schedule, and corrupt $\mathbf{X}_0$ to $\mathbf{Z}_t$ as in Eq. equation 2. The total loss is a weighted sum of three components:

- $\mathcal{L}_{\text{seq}}$: A cross-entropy loss that directly optimizes nucleotide accuracy.
- $\mathcal{L}_{\text{bind}}$: A mean squared error loss that injects residue-level interaction evidence as a soft structural prior.
- $\mathcal{L}_{\text{int}}$: A binary cross-entropy loss that encourages ligand–RNA compatibility.

$$\mathcal{L} = \lambda_{\text{seq}} \underbrace{\frac{1}{BS} \sum_{b,s} \text{CE}\left(\text{softmax}(\mathbf{Z}_{b,s,:}),\, y_{b,s}^{\text{nt}}\right)}_{\mathcal{L}_{\text{seq}}} + \lambda_{\text{bind}} \underbrace{\frac{1}{BS} \sum_{b,s} \left(\hat{y}_{b,s}^{\text{bind}} - y_{b,s}^{\text{bind}}\right)^2}_{\mathcal{L}_{\text{bind}}}$$

$$+ \lambda_{\text{int}} \underbrace{\frac{1}{B} \sum_{b} \text{BCEWithLogits}\left(\hat{y}_b^{\text{int}},\, y_b^{\text{int}}\right)}_{\mathcal{L}_{\text{int}}}. \qquad (12)$$

$y_{b,s}^{\text{nt}} \in \{A, C, G, U\}$ is the target nucleotide, $y_{b,s}^{\text{bind}} \in \{0, 1\}$ is the binding label, and $y_b^{\text{int}} \in \{0, 1\}$ supervises global compatibility (positives when structural conditioning is present). Optionally, an EMA regularizer $\lambda_{\text{EMA}} \|\boldsymbol{\theta} - \bar{\boldsymbol{\theta}}\|_2^2$ with $\boldsymbol{\theta} \in \{\boldsymbol{\gamma}, \boldsymbol{\beta}\}$ can be added to stabilize FiLM at test time.

## 4 EXPERIMENTS

Full details of our experimental setup, including software versions, hyperparameters, and data processing, are provided in Appendix A.4.

## 4.1 DATASET, BASELINES, AND ABLATIONS

Our primary training set contains 1,200 experimental RNA-ligand complexes, with a held-out test set of 192 complexes. We also create an augmented training set by including 1,200 high-confidence

structures generated via AlphaFold3 (Jumper et al., 2021; Abramson et al., 2024). We benchmark LiRDSC against three SOTA models: **RDesign** (Tan et al., 2023), **gRNAde** (Joshi & Lio, 2024), and a fine-tuned version of **RiboDiffusion** (Huang et al., 2024). To validate each component's contribution, we conduct comprehensive ablation studies. These include variants that: (1) remove RNA sequence context from ligand conditioning (**w/o RNA Context**); (2) remove ligand information entirely (**w/o Ligand**); (3) remove all structural input (**w/o Structure**); (4) replace the lightweight FiLM conditioner with a standard **Cross-Attention** mechanism (**w/ Cross-Attention**); (5) trained LiRDSC on **Unfiltered Dataset** to evaluate our confidence-guided data curation strategy towards the *in silico* data (**w/ Unfiltered Data**); (6) trained LiRDSC on the **sub-dataset** which only includes 1,200 high-quality, ground-truth RNA-ligand structures (**w/o *in silico* HCS**).

## 4.2 IMPLEMENTATION AND EVALUATION

**Setup** We use **ChemBERTa** (Chithrananda et al., 2020) and **RNA-FM** (Chen et al., 2022) for ligand and sequence embeddings, respectively, and initialize the GVP encoder with pretrained weights from RiboDiffusion's dataset. The model is trained for 10 epochs using the AdamW optimizer with a learning rate of $10^{-4}$ on a single NVIDIA A100 GPU. Inference is performed via a 10-step iterative diffusive decoding process.

**Evaluation Metrics** We assess performance across multiple levels. (1) **Sequence-Level:** Recovery Rate and Sequence Similarity. (2) **Secondary Structure:** Macro F1-score using predictions from `RNA-FM` (Chen et al., 2022) and `RNAfold` (Hofacker et al., 1994; Lorenz et al., 2011). (3) **Tertiary Structure:** TM-Score and RMSD after inverse folding with **RhoFold** (Shen et al., 2024), **Protenix** (Team et al., 2025) and **AlphaFold3** (Jumper et al., 2021; Abramson et al., 2024). (4) **Binding-Level:** A Binding Score from RNAsmol (Ma et al., 2025) and the Predicted Aligned Error (pAE) from AlphaFold3.

## 5 RESULTS AND ANALYSIS

### 5.1 MAIN RESULTS

LiRDSC establishes a new SOTA in ligand-conditioned RNA inverse folding, demonstrating superior performance across all evaluated metrics (Table 1 and Table 2). Crucially, LiRDSC addresses a more complex and functionally relevant task than the baseline models. While baselines primarily perform general RNA inverse folding (a structure-to-sequence problem), LiRDSC executes ligand-conditioned design, utilizing both structural and small-molecule information as input.

LiRDSC achieves a recovery rate of 60.60%, significantly outperforming the next best model, RiboDiffusion (47.60%). This high recovery is accompanied by a sequence similarity of 58.60%. This metric offers a nuanced insight: while a high value is expected with high recovery, the fact that it is not perfectly aligned with the recovery suggests that LiRDSC generates diverse yet viable sequences. Furthermore, the designed sequences demonstrate excellent secondary structure fidelity, achieving the highest Macro F1-scores on both RNA-FM (84.95%) and RNAfold (87.77%) predictions.

| Methods | Sequence Metrics (↑) | | Secondary Structure (F1) (↑) | |
|---|---|---|---|---|
| | **Recovery Rate (%)** | **Similarity (%)** | **RNA-FM (%)** | **RNAfold (%)** |
| RDesign | $32.78 \pm 1.40$ | $37.13 \pm 0.14$ | $72.55 \pm 0.00$ | $71.20 \pm 0.00$ |
| gRNAde | $36.62 \pm 1.60$ | $44.24 \pm 0.25$ | $75.23 \pm 0.00$ | $73.06 \pm 0.00$ |
| RiboDiffusion | $\underline{47.60 \pm 1.90}$ | $45.40 \pm 0.70$ | $\underline{82.00 \pm 2.64}$ | $\underline{81.55 \pm 2.11}$ |
| **LiRDSC (Ours)** | $\mathbf{60.60 \pm 2.00}$ | $58.60 \pm 1.80$ | $\mathbf{84.95 \pm 0.92}$ | $\mathbf{87.77 \pm 7.59}$ |

Table 1: **Sequence and secondary structure performance.** Macro F1 scores are from RNA-FM and RNAfold. Higher values (↑) are better. Best results are in **bold**; second-best are underlined.

For tertiary structure validation (Table 2), sequences designed by LiRDSC consistently show a superior ability to fold into the target conformation across multiple inverse folding tools. With RhoFold, our method yields the lowest (best) RMSD of 6.6049 Å, and with Protenix, it achieves the highest

| Methods | Folding Tool | lDDT (↑) | RMSD (Å) (↓) | TM-score (↑) |
|---------|------|------|------|------|
| RDesign | Protenix | 0.3106 | 11.3290 | 0.2358 |
| | RhoFold | 0.2782 | 12.2897 | 0.2041 |
| | AlphaFold3 | 0.2117 | 13.3680 | 0.2300 |
| gRNAde | Protenix | 0.3409 | 8.8081 | 0.2886 |
| | RhoFold | 0.3124 | 8.5394 | **0.2761** |
| | AlphaFold3 | 0.2568 | 7.4515 | **0.3279** |
| RiboDiffusion | Protenix | 0.3454 | 10.7844 | 0.2615 |
| | RhoFold | 0.3542 | 8.3028 | 0.2528 |
| | AlphaFold3 | 0.2677 | 9.7315 | 0.2822 |
| **LiRDSC (Ours)** | Protenix | **0.3976** | **8.6308** | **0.2987** |
| | RhoFold | **0.3867** | **6.6049** | 0.2652 |
| | AlphaFold3 | **0.2945** | 7.4048 | 0.3202 |

Table 2: Tertiary structure validation of designed sequences. Lower RMSD (↓) is better; higher lDDT/TM-score (↑) is better. Best results are in bold, second-best are underlined.

lDDT (0.3976) and a leading TM-score (0.2987). The inclusion of AlphaFold3 as a folding tool further reinforces these findings; LiRDSC's designs again produce the lowest RMSD (7.4048 Å) and the highest lDDT (0.2945) among all methods. Notably, achieving the best RMSD with both Rho-Fold and AlphaFold3 underscores our model's particular strength in generating sequences that fold into geometrically precise structures, even when TM-scores are competitive. These results indicate that LiRDSC generates sequences poised to adopt the desired structure, ligand-compatible fold.

## 5.2 ADDITIONAL EVALUATION

In order to ensure a solid and convincing evaluation. We continue to validate our framework on a rigorous data splitting setup, namely CD-HIT (Li & Godzik, 2006). CD-HIT is a very widely used program for clustering and comparing protein or nucleotide sequences. It reduces sequence redundancy and improves the performance of other sequence analyses by a fast program simultaneously.

**CD-HIT Cluster-Based Splitting**: Specifically, we applied CD-HIT to cluster 1,392 RNA-ligand complexes into 140 non-overlapping clusters. We then performed a cluster-wise split into training, validation, and test sets (**8:1:1**), ensuring that no sequence in the test set shares a cluster with any training sequence. This protocol explicitly removes sequence-level redundancy and simulates a more realistic generalization setting. As shown in Table 3, **LIRDSC** consistently outperforms all base-

| Method | Sequence Metrics (↑) | |
|--------|------|------|
| | Recovery Rate (%) | Similarity (%) |
| RDesign | $29.41 \pm 0.10$ | $34.35 \pm 0.10$ |
| gRNAde | $31.93 \pm 0.25$ | $41.52 \pm 1.60$ |
| RiboDiffusion | $57.26 \pm 1.20$ | $53.87 \pm 0.20$ |
| **LiRDSC (Ours)** | **$58.01 \pm 0.25$** | **$55.16 \pm 0.80$** |

Table 3: **Sequence performance on dataset splitted by CD-HIT.** Higher values (↑) are better. Best results are in **bold**; second-best are underlined.

line methods on clustered data splits created using **CD-HIT**, demonstrating strong generalization across structurally diverse complexes. Notably, in the absence of AlphaFold3-generated data augmentation, **RiboDiffusion** exhibits a marked performance improvement, highlighting its reliance on synthetic training data, whereas **LiRDSC** remains robust across all evaluation settings. This robustness is further validated on **RNA3DB** (Szikszai et al., 2024), another benchmark which provides a challenging, family-aware split of experimentally validated RNA-ligand complexes. Using RNA3DB's predefined training and test sets (**723 (training) / 220 (test) entries**), LiRDSC achieves

an overall **Recovery Rate** of **69.60%**, and an overall **Sequence Similarity** of **68.30%**, confirming its effectiveness in generalizing to unseen RNA families.

## 5.3 QUALITATIVE CASE STUDIES

The qualitative results provide compelling visual and quantitative evidence of LiRDSC's strong performance and advantages. **Figure 2** illustrates a standard inverse folding task (folded by Alphafold3 here) on a short RNA (PDB: 2KXM), where LiRDSC achieves the highest recovery rate (**96.30%**) while maintaining excellent structural alignment, showcasing its fundamental strength. **Figure 7** presents a more complex case (PDB: 2M58), where LiRDSC obtains the lowest global RMSD (**24.83 Å**). This indicates that our model prioritizes geometric accuracy in its designs, which belongs to a critical feature for a valid structural scaffold, even when other metrics like TM-score are comparable to baselines.

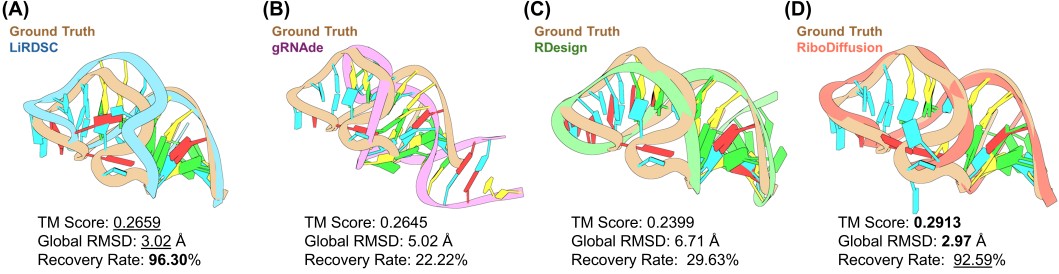

Figure 2: Structure visualization of inverse folding for PDB ID: 2KXM (27 nt). (A) LiRDSC achieves the highest sequence recovery rate (96.30%) and strong structural fidelity (RMSD 3.02 Å). (B–D) Baselines show lower recovery and alignment. On the overall test set, LiRDSC also demonstrates superior functional relevance with a leading Binding Score (84.77) and the lowest pAE of 8.026.

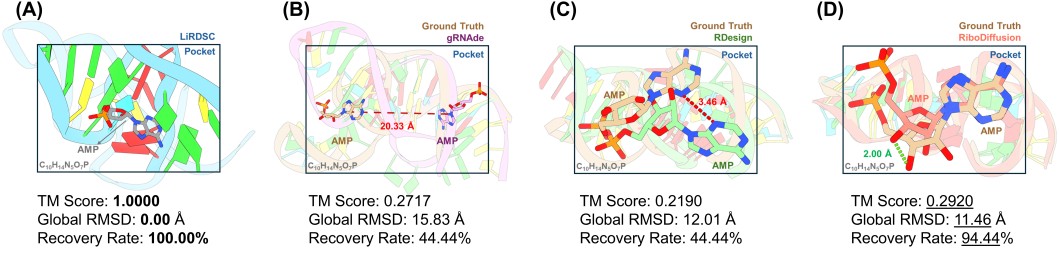

Figure 3: Ligand-conditioned design for PDB ID: 1RAW (36 nt) with ligand AMP. (A) LiRDSC closely recapitulates the RNA fold and ligand-binding pose, achieving 100% recovery, 0.00 Å RMSD, and a 1.0000 TM-score. (B–D) Baselines fail to match this precision. This case highlights LiRDSC's ligand-specificity, further supported by its leading overall Binding Score (84.77) and the lowest pAE (8.026) on the test set.

The true strength of LiRDSC is revealed in the ligand-conditioned case studies. As shown in **Figure 3** (PDB: 1RAW, Ligand: AMP), LiRDSC achieves exceptional results with **100%** sequence recovery, a **0.00 Å** RMSD, and a **1.0000** TM-score. In this compelling example, it closely recapitulates both the RNA fold and the ligand-binding pose, a result not achieved by the baselines in this case, demonstrating a strong capability to design sequences precisely tailored to a specific ligand. This capability is further reinforced in **Figure 8** (PDB: 2AU4, Ligand: GTP). LiRDSC again leads with the highest recovery rate (**97.56%**) and the lowest global RMSD (**10.06 Å**), and the visual comparison clearly shows that its design results in a more accurate ligand placement. The analyses of **Figures 2, 3, 7, and 8** validate that LiRDSC's conditioning mechanisms are highly effective, enabling the design of RNA sequences with high structural fidelity and functional specificity.

## 5.4 ABLATION STUDY

To dissect the contribution of each component, we conduct a comprehensive ablation study (Table 4). The results demonstrate that every module is integral to the model's performance. The most substantial performance drop occurs upon removing structural conditioning (**w/o Structure**), with the recovery rate plummeting to 28.13%. Removing ligand information (**w/o Ligand**) also leads to a significant decline to 47.91%, confirming the efficacy of the LCFC in steering generation toward ligand-specific sequences. The result also shows data-set containing AF3-generated High-Confidence Structures not only provide sequence metrics improvement but also increase in Secondary Structure prediction performance.

| Methods | Sequence Metrics (↑) | | Secondary Structure (F1) (↑) | |
|---|---|---|---|---|
| | Recovery Rate (%) | Similarity (%) | RNA-FM (%) | RNAfold (%) |
| w/o RNA Context | 49.16 | 47.91 | 72.99 | 71.65 |
| w/o Ligand | 47.91 | 46.92 | 70.37 | 69.71 |
| w/o Structure | 28.13 | 17.61 | 67.20 | 67.16 |
| w/ Cross-Attention | 45.19 | 43.78 | 71.49 | 71.55 |
| w/ Unfiltered Data | 45.06 | 45.41 | 74.56 | 72.76 |
| w/o *in silico* HCS | 59.20 | 60.10 | 82.19 | 83.10 |
| **LiRDSC (Ours)** | **60.60** | 58.60 | **84.95** | **87.77** |

Table 4: Ablation study of LiRDSC components. "**HCS**" stands for "**High-Confidence Structure**". Performance of the full model is compared against ablated variants. Best results are in **bold**, second-best are underlined.

## 5.5 BINDING ANALYSIS

To further probe the functional relevance of the designs, we analyze predicted binding metrics (Table 5). LiRDSC achieves a Binding Score of 84.77 from RNAsmol, the highest among all generative models and closest to the ground truth. More strikingly, when evaluating the designed RNA-ligand complexes with AlphaFold3, LiRDSC yields a pAE of 8.026, the lowest of all methods. This dual success is a crucial insight: LiRDSC not only generates sequences predicted to have high binding affinity but also designs them to form structurally confident and stable complexes.

| Methods | Ground Truth | LiRDSC (Ours) | RDesign | gRNAde | RiboDiffusion |
|---|---|---|---|---|---|
| Binding Score | 90.44 | **84.77** | 82.65 | 79.16 | 84.10 |
| pAE (AF3) | – | **8.026** | 11.165 | 8.612 | 8.798 |

Table 5: **Binding analysis of designed sequences.** Comparison of the RNAsmol Binding Score (higher is better) and AlphaFold3 interface pAE (lower is better) for sequences generated by LiRDSC and baselines. Best results are in **bold**, second-best are underlined.

## 6 CONCLUSION AND OUTLOOK

This work introduces LiRDSC, a deep generative framework that uses diffusive conditioning to achieve specific RNA design. It demonstrates SOTA performance in generating diverse and structurally viable candidates, providing a powerful tool for molecular engineering.

Biological experimental validation is required to confirm the binding affinity of the designed sequences. The framework can be further extended to optimize for binding affinity and selectivity against similar ligands, a critical step for creating effective RNA-based biosensors and therapeutics.

## ETHICS STATEMENT

This work targets ligand-conditioned RNA sequence design for benign applications (e.g., biosensing, therapeutics). It uses (i) publicly available macromolecular structures from the RCSB PDB and (ii) *in silico* complexes predicted by an AlphaFold-based server. No human-subjects data or personally identifiable information are involved, and no IRB approval is required.

**Licensing.** PDB records are public-domain releases; predicted structures generated via the AlphaFold-based server are redistributed or reproduced only under the server's non-commercial terms and are clearly labeled in any release.

**Potential dual-use (biosecurity).** While our work targets benign applications (biosensing and therapeutics) and uses only public or *in silico* molecular data, sequence design methods could in principle be repurposed. We therefore (i) transparently document ligand selection and exclusion rules used in our experiments, and (ii) will accompany any artifact release with usage guidelines that discourage harmful applications. No human-subjects or privacy-sensitive data are used.

**Third-party attributions.** We credit and abide by licenses of all third-party baselines and pretrained models. The authors declare no competing interests; funding acknowledgments will appear in the camera-ready as per conference policy.

## REPRODUCIBILITY STATEMENT

We detail all ingredients needed to reproduce our results:

**Setups.** Section 4 and Appendix A.4 specifies software versions, hardware, optimizer settings, batch sizes, noise schedules, and fixed random seeds; we report mean $\pm$ std over multiple runs.

**Training and evaluation.** The paper and appendix provide data curation rules (PDB filters, single-chain constraint, distance cutoff), augmentation criteria (selection by confidence scores), and exact evaluation metrics and libraries (Recovery/Similarity, secondary-structure F1, TM-score/RMSD/lDDT).

**Baselines.** We list the official implementations and release tags used and state any deviations from defaults.

**Artifact release plan.** Upon acceptance, we will release source code, configs, and a data card. For AlphaFold-server–derived structures, we will either distribute metadata and reproduction scripts under the server's non-commercial terms or label any shared coordinates accordingly.

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

# A APPENDIX

## A.1 USE OF LARGE LANGUAGE MODELS (LLMS)

In accordance with ICLR 2026 policies, we disclose that a general-purpose LLM was used *only* for grammar and wording edits of this manuscript. LLMs were not used for research ideation, data collection or generation, code implementation, experimental design, or result analysis. No non-public data were provided to LLM tools. All content was verified by the authors, who take full responsibility for any errors. The same disclosure is provided in the submission form.

## A.2 RELATED WORKS

### A.2.1 RNA DESIGN

The design of RNA has rapidly progressed from classical combinatorial search and physics-based methods (Laganà et al., 2014) to data-driven deep learning approaches. Modern methods increasingly leverage tertiary structural information, primarily through Graph Neural Networks (GNNs) that process molecular geometry (Gilmer et al., 2017; Atz et al., 2021). To correctly handle the geometric degrees of freedom, SE(3)-equivariant architectures such as Geometric Vector Perceptron (GVP) have become instrumental (Fuchs et al., 2020; Jing et al., 2020), leading to powerful structure-based designers like `gRNAde` (Joshi & Lio, 2024) and `RDesign` (Tan et al., 2023). Most recently, diffusion probabilistic models have set a new SOTA in generative structural biology for both proteins (Watson et al., 2022; Ingraham et al., 2023) and RNA, as exemplified by `RiboDiffusion` (Huang et al., 2024). Our work builds on this paradigm by developing a diffusion-based structural encoder specifically for the ligand-conditioned setting.

### A.2.2 LIGAND-CONDITIONED GENERATION AND BINDING PREDICTION

While predicting RNA-ligand binding is a well-studied problem addressed by both classical docking (Trott & Olson, 2010) and deep learning-based scoring functions (Zeng et al., 2024; Yan & Zhu, 2020), the inverse task of designing an RNA for a specific ligand is far more challenging and less explored. A primary difficulty is ensuring the generative model produces sequences that are chemically and sterically compatible with the ligand, rather than collapsing to generic, non-binding structures. This problem of conditioned generation is an active area of research in protein design (Anand et al., 2022; Strokach & Kim, 2022), but remains nascent for RNA. Effectively steering the generative process requires powerful conditioning mechanisms, such as cross-attention (Vaswani et al., 2017) or feature-wise modulation layers like FiLM (Perez et al., 2018). The recent success of AlphaFold3 in modeling RNA-ligand complexes further enables new opportunities for training such models (Abramson et al., 2024). LiRDSC directly confronts this challenge by using a Ligand-Contextual FiLM Conditioner to enforce ligand specificity throughout the design process.

## A.3 DATASET STRUCTURAL DIVERSITY ANALYSIS

## A.4 EXPERIMENTS

### A.4.1 DATASET

Our primary dataset consists of 1,200 experimentally validated RNA-ligand complexes for training and a distinct set of 192 complexes for testing and another group of 92 for validating. The train-val-test split is pre-fixed to ensure consistent and reproducible evaluation across all experiments. To study the effect of data augmentation, we also create an extended training set of 2,400 complexes by incorporating an additional 1,200 high-confidence structures generated via AlphaFold3, as detailed in Section 2.

### A.4.2 IMPLEMENTATION DETAILS

**Data and Feature Engineering** The model is trained on RNA-ligand complexes derived from PDB structures. For each complex, we parse the PDB file to generate graph tensors that include node scalars and vectors, positional encodings, and k-Nearest Neighbors (kNN) edges. To generate

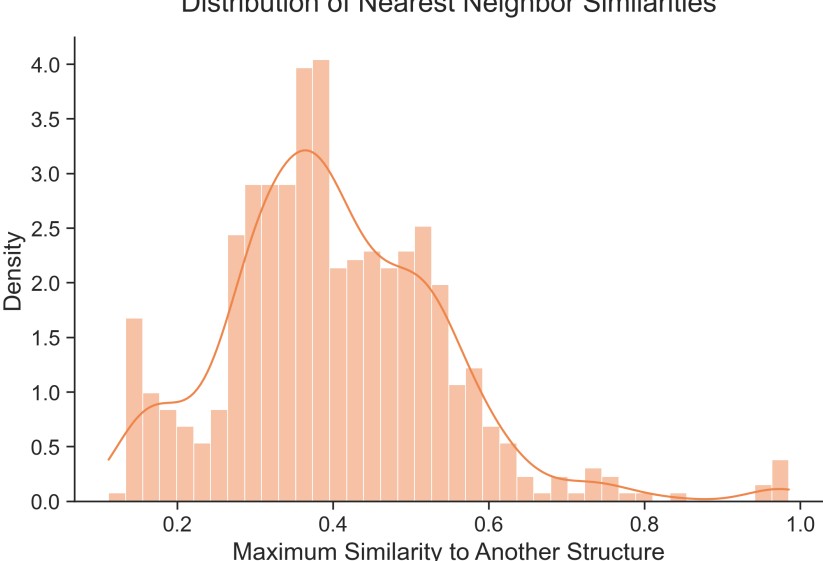

Figure 4: **Distribution of nearest-neighbor structural similarity within the entire dataset.** For each entry, the maximum TM-score against all other structures was computed to quantify its closest structural neighbor. The resulting probability density plot shows that most entries exhibit a nearest-neighbor similarity below 0.6, with a strong concentration between 0.3 and 0.5. This wide spread of similarity scores indicates substantial structural diversity across the dataset. Importantly, the lack of high-similarity peaks near 1.0 suggests no obvious redundancy or data leakage, confirming that structurally similar entries are rare and that each example contributes uniquely to the dataset.

informative input representations, we employ two pretrained models as featurizers. We use **Chem-BERTa** (Chithrananda et al., 2020) to produce precomputed embeddings for the ligand ($\mathbf{h}^{\text{lig}}$) from its SMILES string and **RNA-FM** (Chen et al., 2022; Shen et al., 2024) to generate embeddings for the RNA sequence ($\mathbf{H}^{\text{seq}}$). Residue-wise binding masks ($\mathbf{y}^{\text{bind}}$) are also generated and aligned to the RNA residue indices. For the structural encoding module, the weights of the Geometric Vector Perceptron (GVP) component are initialized from a model pretrained on RiboDiffusion's dataset, allowing it to effectively capture tertiary structure geometric features.

**Training Process and Optimization**    The training strategy is designed to explicitly align sequence prediction with structural denoising. By minimizing $\mathcal{L}_{\text{seq}}$ over a random distribution of noise levels $t$, the model learns to make predictions across a spectrum of structural perturbations, as described in Eq. equation 2. This strong coupling between the sequence decoder and the structure encoder encourages the design of RNA sequences that are compatible with a given structure. The model is implemented in Python 3.11.9, utilizing libraries such as PyTorch 2.6.0 and Transformers 4.54.0. We optimize the model parameters $\psi$ using the AdamW optimizer with a learning rate of $10^{-4}$, which is decayed during training. Training is conducted for 10 epochs with a batch size of 1 on a single NVIDIA A100 GPU (80 GB), where one epoch on the 1,200-complex dataset completes in approximately 90 seconds. To improve stability and generalization, we maintain an exponential moving average (EMA) of the model's trainable parameters, and use these EMA-averaged weights for evaluation.

**Sampling and Inference**    During evaluation, the model generates RNA sequences via the diffusive decoding process. First, we freeze the parameters of the GVPtrans structural encoder and the LCFC EMA buffers for $(\bar{\gamma}, \bar{\beta})$. We set the noise level from $t = -10$ to $t = 10$use the noise-injected structural input for obtaining the structural embedding $\mathbf{H}^{\text{struct}}$. The sequence-agnostic ligand representation is then computed as $\mathbf{h}^* = \bar{\gamma} \odot \mathbf{h} + \bar{\beta}$. Finally, the model generates a nucleotide sequence through an iterative diffusive decoding process, which requires 32 seconds for an optimal 10 steps. To explore the design space, multiple stochastic runs can be performed to generate a pool of can-

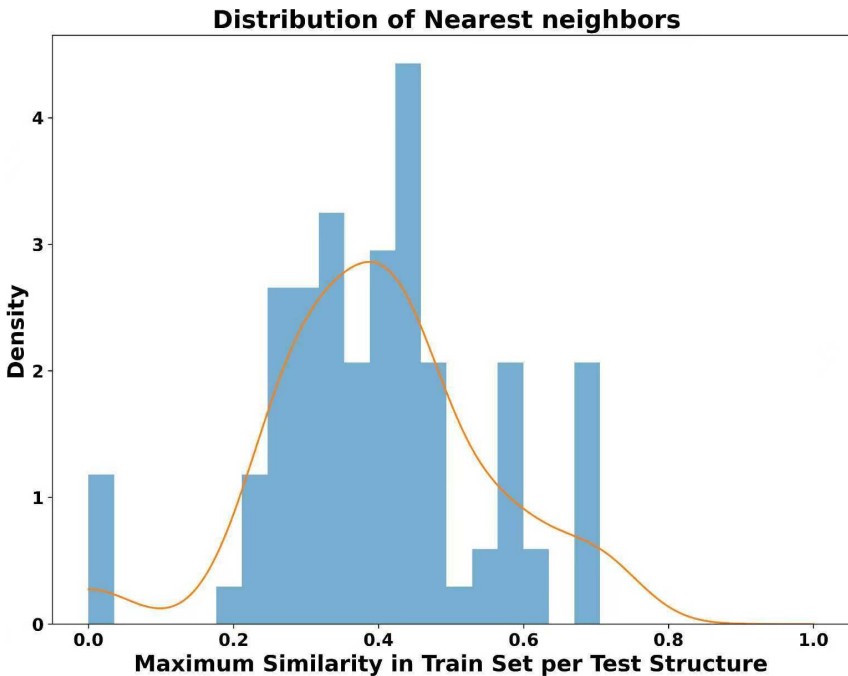

Figure 5: **Distribution of nearest-neighbor structural similarity between test and training sets.** For each RNA–ligand complex in the test set, we compute its maximum TM-score against all training set structures to quantify structural proximity. The resulting distribution shows that most test entries have no close structural analogs in the training set, with nearest-neighbor TM-scores predominantly below 0.6 and strongly concentrated between 0.3 and 0.5. The absence of high-similarity peaks near 1.0 indicates minimal structural redundancy and confirms that the train/test split reflects a realistic generalization setting.

didate sequences, which can then be ranked based on predicted binding propensities ($\hat{y}^{\text{bind}}$) and the global interaction score ($\hat{y}^{\text{int}}$).

### A.4.3 EVALUATION METRICS

To assess model performance, we use metrics evaluating both sequence and structure quality.

**Sequence-Level Metrics** Following standard practice in inverse folding, we use **Recovery Rate** as the primary metric, which measures the percentage of correctly predicted nucleotides compared to the ground-truth sequence. In addition, we compute a **Sequence Similarity** score using local sequence alignment (Biopython's 'pairwise2.localxx' algorithm). This score is calculated as the ratio of identical characters to the total alignment length in the best local alignment, providing a normalized measure of local identity.

**Secondary Structure-Level Metrics** For secondary structure evaluation, we predicted the secondary structure for each designed sequence using two tools, `RNA-FM` (Chen et al., 2022; Shen et al., 2024) and `RNAfold` (Hofacker et al., 1994; Lorenz et al., 2011). Performance was quantified by the **Macro F1-score** against the ground-truth secondary structure.

**Tertiary Structure-Level Metrics** To evaluate whether the designed sequences can fold into the desired tertiary structures, we perform inverse folding using two tools, **RhoFold** (Shen et al., 2024) and **Protenix** (Team et al., 2025). We compare the predicted structures against the native structures using **TM-Score** and **Root-Mean-Square Deviation (RMSD)**. Higher TM-Score and lower RMSD indicate a closer match to the target structure.

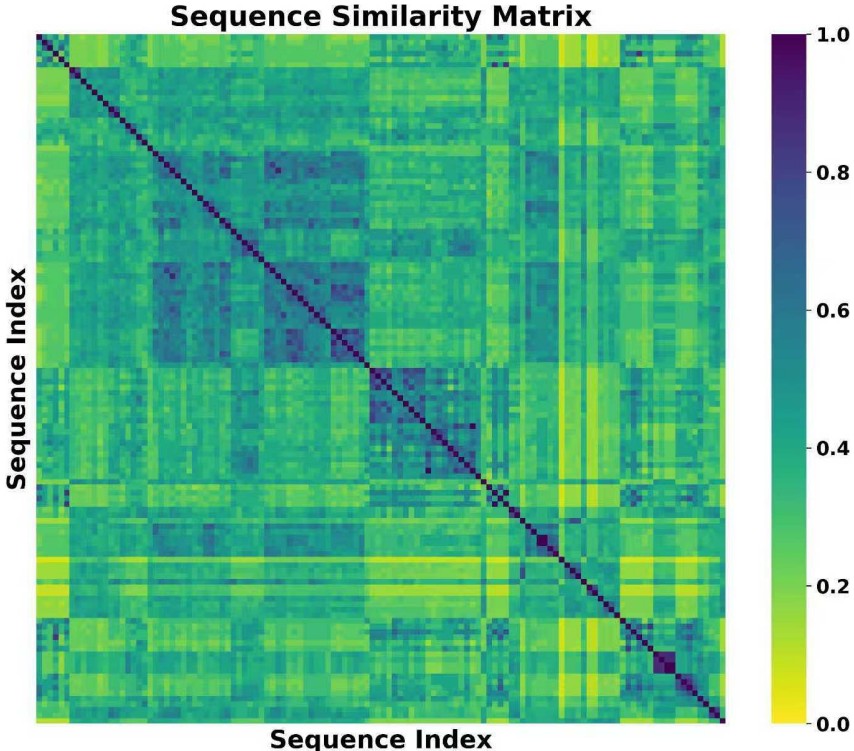

Figure 6: **Pairwise sequence similarity matrix of RNA sequences generated by LiRDSC.** Each cell represents the normalized local alignment score between a pair of generated RNA sequences, with values ranging from 0 (no similarity) to 1 (identical). The low off-diagonal similarity values indicate that LiRDSC produces a highly diverse set of sequences, despite being conditioned on structurally similar inputs. This sequence-level diversity suggests that the model avoids mode collapse and generates multiple viable, ligand-compatible sequence variants.

**Binding-Level Metrics** To evaluate the structural confidence of the designed RNA-ligand complexes, we utilized AlphaFold3 to calculate the pAE at the molecule-interface level. Due to the computational cost of this analysis, the pAE was evaluated on a randomly sampled subset of 60 complexes from the test set. A lower pAE value indicates a more reliable prediction of the interaction interface.

### A.4.4 BASELINES AND ABLATION STUDIES

We compare LiRDSC against external SOTA models and internal ablations of our framework to validate each component's contribution.

**External Baselines.** We benchmark against three models: **RDesign** (Tan et al., 2023), **gR-NAde** (Joshi & Lio, 2024), and **RiboDiffusion** (Huang et al., 2024). To ensure a fair comparison, we fine-tune the official RiboDiffusion model on our training dataset. For RDesign and gRNAde, we evaluate their officially released pretrained models. Importantly, in all baseline evaluations, we provide the same RNA–ligand complex structures as input, ensuring that each method is conditioned on the corresponding ligand. All baseline experiments are conducted using their official implementations.

**Ablation Models.** To isolate the contribution of each component, we define the following ablated variants of LiRDSC: (1) **w/o RNA Context:** A variant where the ligand embedding is not conditioned on the RNA sequence context. (2) **w/o Ligand:** A variant where the ligand embedding is removed entirely from the conditioning process. (3) **w/o Structure:** A variant where the GVPtrans structural embeddings are removed from the model. (4) **w/ Cross-Attention:** A variant that replaces the lightweight FiLM conditioner with a standard Transformer cross-attention mechanism. (5) **w/**

**Unfiltered Data:** A variant trained on a larger, but loosely filtered set of synthetic structures to evaluate the impact of our confidence-guided data curation strategy.

## A.5 CASE STUDY

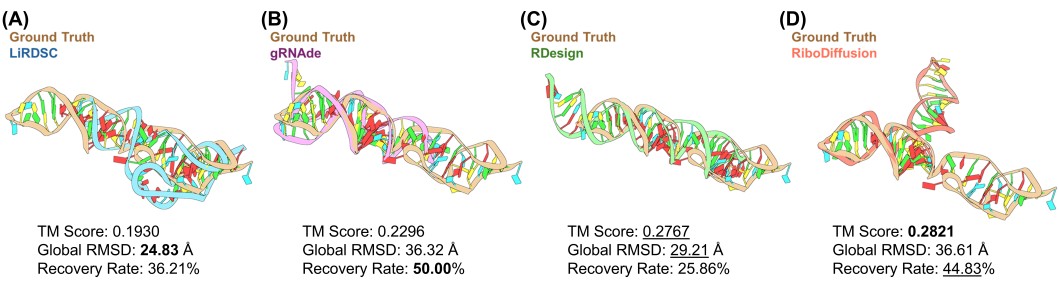

Figure 7: Structure visualization of a designed RNA sequence conditioned on the ground truth RNA structure (PDB ID: 2M58, 58 nt). (**A**) LiRDSC achieves the lowest global RMSD of **24.83** Å, reflecting strong structural alignment with the ground truth. (**B–D**) Comparisons with baseline methods—gRNAde, RDesign, and RiboDiffusion respectively—show that while LiRDSC achieves the best global RMSD (↓), other methods yield higher TM scores (↑) and recovery rates (↑). These results demonstrate that LiRDSC generates geometrically accurate structures, reinforcing its strength in RNA design conditioned on known structures. Best results are in **bold**, and second-best are underlined. For the global RMSD, lower is better (↓), while for recovery rate and TM-score, higher is better (↑).

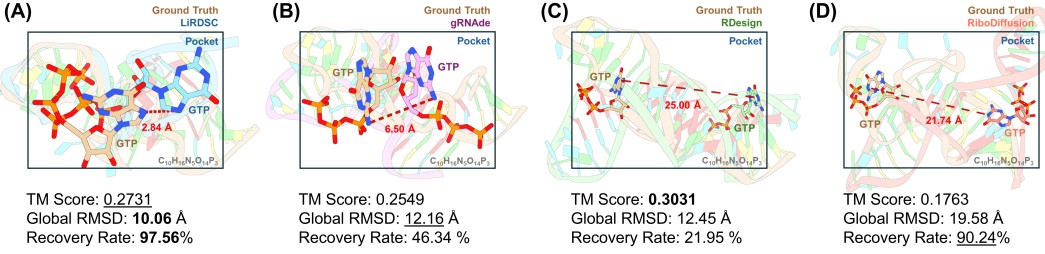

Figure 8: Structure visualization of a designed RNA sequence conditioned on the ground truth RNA structure (PDB ID: 2AU4, 41 nt), which experimentally binds to the unique ligand GTP. For each model, the generated RNA sequence was folded, docked with GTP, and then structurally aligned to the ground truth complex. (**A**) LiRDSC achieves the highest recovery rate of **97.56**% and the lowest global RMSD of **10.06** Å, indicating high structural fidelity, a key prerequisite for function. It also attains the second-best TM score. (**B–D**) Baseline methods—gRNAde, RDesign, and RiboDiffusion respectively—exhibit lower recovery rates and less accurate ligand placements, with higher global RMSD and TM-score variability. These results further emphasize LiRDSC's effectiveness in generating ligand-compatible RNA structures. Best results are in **bold**, and second-best are underlined. For global RMSD, lower is better (↓), while for recovery rate and TM-score, higher is better (↑).

