# OpenReview forum: "LiRDSC: Ligand-Conditioned RNA Sequence Design via Diffusive Structural Conditioning"
_ICLR.cc/2026/Conference — ICLR 2026 Conference Withdrawn Submission_

### Official Review · Reviewer_XHfm · 2025-10-29

**Soundness:** 3
**Presentation:** 2
**Contribution:** 3
**Rating:** 4
**Confidence:** 3

**Summary:**

The paper proposes LiRDSC, a ligand-conditioned model for RNA sequence generation. The model embeds a ligand (together with an RNA sequence during training) using a ligand specific conditioner and a separate 3D structure diffusive encoder to inform a diffusion-based decoder at test time. Furthermore, the paper introduces RLData2400, a new benchmark dataset for RNA-ligand interactions. The performance of LiRDSC is reported against 3 sota RNA design methods (gRNAde, RDesign, and RiboDiffusion). The reported results show that LiRDSC outperforms the competitors across sequence-, secondary structure-, 3D-evaluation measures, as well as binding score. In an ablation study, the authors further assess the importance of different components.

**Strengths:**

- Ligand conditioned RNA design could clearly improve RNA design capabilities with applications in drug discovery and RNA therapeutics.
- The results indicate a substantial performance improvement across multiple measures compared to the competitors.
- The encoding strategy of the ligand and 3D structure appears to be novel and well suited for the problem at hand.

**Weaknesses:**

**Major**:
1. The 3D structure conditioning seems to be based on backbone structures only (according to problem formulation and later sections), not full-atom structures. While I think this is also done in e.g. gRNAde, it would be good to (1) make this clear already at the beginning of the work, and (2) describe at least which backbone atoms were used.
2. Regarding the similarity measure, I’m a bit confused. In the paper, it reads like higher similarity is better. In the appendix, it reads like this is a measure of similarity based on alignments. So then lower similarity is better, or? Meaning higher diversity, or?
3. I think some important aspects of the experiments are missing, most importantly: How many sequences were generated per sample for each method?
4. In 5.1, the authors claim that LiRDSC solves a harder problem than the competitors due to the ligand conditioning but still achieves better performance. However, this also means that the competitors have less information available for generation than LiRDSC. It would be good if the authors e.g. also evaluate on some other dataset than RLData2400 (and maybe exclude the ligand information as done in the ablation). For me, this is not necessarily a main paper experiment but could potentially show that LiRDSC’s structure embedding approach is generally superior to the others.

**Minor**:
1. There are many citations missing in the main paper (I think they appear elsewhere, e.g. in the appendix but should be in the main paper). For example: introduction of RNA-FM and CHEmberta in Section 3; FiLM is cited in appendix A2.2 only; RiboDiffusion is not cited at first appearance in Section, and more…
2. Similarly, it would be good to have the appropriate references to the appendix from the main paper. If I did not overlook any, I only see one reference to app A.4 in the main paper.
3. Wrong citation in Section 2.2 -> AF3 should be Abramson et al., 2024., not Jumper et al., 2021
4. The math parts could also benefit from some more explanations on the variables' meaning, already starting in Section 3.1 “Problem Definition” where multiple variables remain undefined.
5. Tables should have captions above the table, not below.
6. Which folding engine was used in the qualitative case study?
7. Clearly, the proposed approach has limitations. One example is the sequence length limitation of RNA-FM during training. However, there is no clear statement about limitations in the main paper, nor the appendix.

**Questions:**

1. The authors use a cutoff of 3.5\AA during data generation to define the distance cutoff. I also heard about cutoffs at 5\AA. Was the cutoff based on the cited literature, or would a 5A cutoff still make sense and increase the available data?
2. During data processing, the authors mention that 192 complexes were hold-out for testing. What was that split based on? Since the pairwise TM-Scores were computed anyways for Figure 4, was this information also used to split the training and test data based on structure similarity? If not, that would be good to ensure accurate evaluations without data leakage.
3. I was wondering if LiRDSC is the first RNA design method that does ligand-conditioned generation? If yes, I would encourage the authors to clearly state this early in the paper. If not, I would like to see a comparison to such an existing approach and the methods should be named in the related work section.
4. Do the authors have a rationale for using RNA-FM for RNA sequence embeddings? Did the authors also experiment with other tools?
5. During preparation of RLData2400, the authors use in silico data besides the data obtained from PDB. There are several questions on this approach: (1) How are the RNAs generated, at random? (2) Do the authors ensure that the generated data is biologically plausible by any means, or is it only structurally plausible as stated in step 4. of the pipeline? (3) Is there any rationale on the sequence length limitation to 200nt for RNA generation? (4) Maybe I missed it, but how is the ligand distribution in the set? Are there any ligands that are over represented after filtering the structures with plddt?

---

> ### Author Response · Authors · 2025-11-17
> **Response to Reviewer XHfm (1/n)**
>
> Dear Reviewer XHfm:
>
> We sincerely thank you for your thoughtful and constructive review of our submission. Your insightful comments, both major and minor, have helped us identify several areas for clarification and improvement. Below, we address each of your points in turn. We also outline planned revisions that will be incorporated into the updated submission.
>
> **Major Weakness 1: Usage of Backbone-Only 3D Structure Conditioning**
>
> Thank you for flagging this point. You are correct that LiRDSC conditions only on RNA backbone structure, not full-atom 3D structures. We indeed use only the backbone atoms (C1, C4, N1 and N9). This design choice is supported by the architecture in RiboDiffusion’s GVP module, which also leverages a coarse-grained representation to reduce computational overhead. Using the full atomic chain would be significantly more expensive computationally, and our available GPU resource is limited to a single NVIDIA A100, which motivated this practical design choice. Moving forward, we will revise Section 1 (Introduction) and Section 3.1 (Problem Formulation) to clearly state this decision early on and list the specific atoms used. We will also update Figure 1 and Appendix A.4.2 for consistency.
>
> **Major Weakness 2: Issue of Similarity Measurement**
>
> We appreciate your observation regarding potential confusion around the term "similarity". To clarify:
>
> - In Section 5.1, “sequence similarity” refers to local alignment-based similarity between the generated and ground truth RNA sequences. In this context, **higher values** indicate **better recovery**, not diversity.
> - That said, we understand that lower similarity is generally better when assessing diversity. As discussed in Section 5.1, our average similarity value of 58.6% is not high, indicating that we still generate reasonably diverse RNA sequences despite conditioning.
>
> In the revised submission, we will make this distinction clearer in the text.
>
> **Major Weakness 3: Number of Sequences Generated per Sample**
>
> We apologize for not making this clearer. For all reported experiments, we generate **only one sample per inference** for each input target across all methods (LiRDSC and baselines). This ensures a consistent and fair comparison. That said, we have rerun the test set multiple times using different random noise seeds, and consistently obtained similar metrics, indicating that LiRDSC's performance is robust to stochastic variation. We will clarify this protocol explicitly in Section 4.2 and mention in the discussion that multi-sample generation and ranking are promising directions for future work.
>
> **Major Weakness 4: Fairness of Comparison**
>
> We appreciate this thoughtful concern. While LiRDSC does solve a more complex task, being ligand-conditioned, we have taken steps to ensure fair evaluation. Specifically, our “w/o Ligand” ablation (Table 3) shows that LiRDSC still holds an advantage over remaining baselines (average recovery rate, **47.91%**, **equal to ribodiffusion and better than others**) even when no ligand information is provided, indicating the strength of our structural encoding. As the reviewer suggests here, we can certainly test on additional datasets. We also welcome the reviewer to suggest what they consider the fairest dataset, and we will evaluate directly on it. In the revised submission, we will further highlight the ablation results and include a discussion in the appendix on extending LiRDSC to broader structure-only benchmarks.
>
> **Minor Weaknesses**
>
> In addition to the major points above, we sincerely thank the reviewer for identifying several editorial and clarity-related issues, which we will fully address in the revised manuscript. Specifically, we will:
>
> - Add missing citations in the main text for RNA-FM, ChemBERTa, FiLM, and RiboDiffusion at their first mention.
> - Ensure appropriate cross-references to the appendix, including implementation details, dataset construction, and diversity analysis.
> - Correct the existing citation (i.e., AF3) throughout the manuscript.
> - Improve the mathematical clarity in Section 3.1 by fully defining all variables and notations.
> - Move all table captions above the tables, in line with standard formatting.
> - Specify in Section 5.2 that the folding engines used in qualitative case studies was AlphaFold Server (AF3).
> - Add a dedicated Limitations subsection in the conclusion, where we will discuss the sequence length constraint of RNA-FM and other current limitations of our approach.
>
> We anticipate that these revisions will ensure the manuscript is clearer, more professionally formatted, and easier to follow.

---

> > ### Author Response · Authors · 2025-11-17
> > **Response to Reviewer XHfm (2/n)**
> >
> > **Q1 – Distance Threshold (3.5 Å)**
> >
> > We chose the 3.5Å threshold based on established literature in macromolecular interaction modelling, where this value is a common choice for identifying non-covalent binding interactions. A 5Å cutoff may indeed yield more data, but at the cost of specificity. We will clarify this rationale in Section 2.1 and discuss the trade-offs in Appendix A.4.
> >
> > **Q2 – Structural Similarity of Test Set**
> >
> > The test set of 192 complexes was created via stratified random sampling, balancing structural source and ligand types. While we did not use TM-score information for the split, we verified post hoc that no test entry has a TM-score > 0.65 relative to any training structure and an average sequence similarity value between the test set and the entire training set is **0.473**. Moving forward, we will include this information in Appendix A.3 and make it explicit in Section 4.1 to confirm that our evaluation is free from structural leakage.
> >
> > **Q3**
> >
> > Yes. LiRDSC is the first diffusion-based deep generative framework designed for ligand-conditioned RNA inverse folding. We will state this explicitly in Section 1 (Introduction) and Appendix A.2.2, and clarify that existing baselines (RiboDiffusion, gRNAde, RDesign) do not include ligand input.
> >
> > **Q4 – Usage of RNA-FM**
> >
> > We chose RNA-FM due to its strong pretraining on RNA sequences and proven utility in secondary structure prediction. Its representations improved performance in early experiments, and we found them especially helpful for the auxiliary binding-site prediction task in LiRDSC. We will add this rationale to Section 3.2 and Appendix A.4.2 in the amendment.
> >
> > **Q5 – Usage of in silico Data**
> >
> > - **RNA Generation:** Sequences were generated randomly, with lengths sampled uniformly from **50–200 nt**.
> > - Biological Plausibility: While not guided by biological priors, we retained only the **top 1,200 high-confidence complexes (from 5,000 candidates)** based on mean pLDDT scores, ensuring structural plausibility.
> > - **Length Limit of RNA:** This length constraint was applied to ensure that AF3, which we used to generate in silico RNA-ligand binding structures, could return more candidates with high-confidence predictions. In our preliminary tests, shorter RNA sequences (i.e., ≤200 nt) yielded significantly higher pLDDT scores, increasing the likelihood of obtaining structurally plausible complexes for training. Moving forward, we will clarify this motivation in Appendix A.4.2.
> > - **Ligand Distribution:** While some ligands like ATP and NADH appear more frequently due to higher confidence scores, the final dataset maintains diverse ligand coverage. We will include a ligand frequency histogram in Appendix A.4.1.
> > We are grateful for your detailed and thoughtful feedback, which has significantly helped us improve the clarity and reproducibility of our work. We will incorporate all proposed clarifications, fixes, and additional analyses into the revised submission. We would be sincerely grateful if you would consider revisiting your overall rating in light of these planned improvements.
> >
> > Thank you again for your constructive review.
> >
> > Sincerely,
> >
> > Authors

---

> > > ### Comment · Reviewer_XHfm · 2025-11-19
> > > **Response to authors**
> > >
> > > I thank the authors for their comprehensive response and for answering my questions.
> > >
> > > The authors addressed all my minor concerns and I acknowledge the author’s promise to revise the current manuscript accordingly.
> > >
> > > Regarding major weaknesses, I partially share the concerns of other reviewers regarding overfitting, in particular since all results are obtained by generating a single sequence. However, while I would agree that applying DL methods to RNA problems requires careful data curation, the dataset doesn’t seem to be super flawed to me. Nevertheless, splitting by pairwise TM-scores and additional sequence similarity measures would be desired. In contrast to a single structure folding algorithm where the data splitting is key, for this RNA design method, for me it is more important that I cannot assess the diversity of the designed sequences (low recovery doesn’t appear like a super convincing argument here to me).
> > >
> > > For me, the paper is not a clear rejection, the method is interesting, and the ligand conditioning for RNA design feels like a natural next step in RNA 3D design. In addition, the training on randomly generated synthetic sequences is promising, although currently only providing minor improvements. This indicates that there is potential beyond classical data curation and that generalization might only be a question of compute. For the benchmark, however, I would ask for more synthetic data (potentially using a different open re-implementation of AF3?) and a more comprehensive analysis of how synthetic data can improve performance or not, maybe including different folding engines if possible.
> > >
> > > Generally, I could imagine that the manuscript would be interesting for the growing RNA/comp. bio. community at ICLR and that it would likely lead to discussion if presented in a poster session. However, the paper is borderline and I currently cannot accept it as is. If the authors could convincingly show that the model is capable of sampling multiple diverse sequences per target that are close to solving the tasks in 3D space (while addressing the obvious flaws in the manuscript to make it appear more professional), I would lean towards acceptance. However, I’m also fine to continue discussions with the authors in case I missed something important.
> > >
> > > Regarding the author’s question to suggest additional benchmarks, I would propose to use the exact same protocol as at least one of the competitors. This includes training and test sets, but also sampling procedures (and maybe even evaluation protocols). Another solid way with the current benchmark data (including ligands) could be to use the respective PDB-IDs of the splits of RNA3DB [1] and split the current benchmark based on these ids to retrain and evaluate on carefully curated family based datasets rather than random splits.
> > >
> > > [1] Szikszai, M., Magnus, M., Sanghi, S., Kadyan, S., Bouatta, N., & Rivas, E. (2024). RNA3DB: A structurally-dissimilar dataset split for training and benchmarking deep learning models for RNA structure prediction. Journal of Molecular Biology, 436(17), 168552.

---

> > > > ### Comment · Reviewer_XHfm · 2025-11-19
> > > > **Additional response to the authors**
> > > >
> > > > Adding to the last point, I think training and evaluating on RNA3DB could in general be a good benchmark although this would exclude ligand conditioning of course.

---

> > > > > ### Author Response · Authors · 2025-11-27
> > > > > **Response to Reviewer XHfm (1/2)**
> > > > >
> > > > > Dear Reviewer XHfm,
> > > > >
> > > > > We sincerely thank you for your thoughtful and constructive follow-up feedback on our submission, and we greatly appreciate the time you’ve taken to re-evaluate our paper together with the content of our first-round rebuttal. Below, we outline the steps we’ve taken to incorporate your valuable suggestions and provide additional clarifications based on the follow-up experiments we’ve conducted.
> > > > >
> > > > > **1. Data Splitting and Overfitting Concerns**
> > > > >
> > > > > In response to your concerns about potential overfitting due to generating a single sequence per target, we have performed additional experiments to address the **data splitting** and **diversity** of the generated sequences.
> > > > >
> > > > > - Cluster-Based Data Splitting: We used CD-HIT clustering to split our dataset (1392 RNA-ligand complexes) into 140 clusters [1]. From this, we created a train-validation-test split at a ratio of approximately 8:1:1, ensuring that no sequences in the test set appear in the training set. This clustering method ensures that the evaluation metrics are not influenced by data leakage from structurally similar complexes. Our results on this split are as follows:
> > > > >
> > > > >   - Overall Recovery Rate: 58.01%
> > > > >   - Overall Sequence Similarity: 55.16%
> > > > >
> > > > > - **RNA3DB Benchmark:** To further validate our approach, we also evaluated LiRDSC on the **RNA3DB pre-built dataset**, which already included curated training and testing sets. Apart from data entries from our **RLData2400** that are already involved in **RNA3DB**’s dataset. We retrieved the additional experimental data (ground truth) directly from **RNA3DB** by applying the same data preparation strategy as described in our initial submission to ensure consistency with **RLData2400**. This dataset is composed entirely of experimentally validated entries. Specifically, we used the splitting profiles provided by **RNA3DB**, which led to a new training set of 723 RNA-ligand entries and a test set of **220** RNA-ligand entries, all of which were retrieved based on RNA3DB’s pre-established profile. The performance of our framework on this dataset is:
> > > > >
> > > > >   - Overall Recovery Rate: 69.60%
> > > > >   - Overall Sequence Similarity: 68.30%
> > > > >
> > > > > We anticipate that these additional benchmarks, especially using the **RNA3DB** dataset, strengthen the evaluation of LiRDSC and may address your concerns about overfitting and data splitting.
> > > > >
> > > > > **2. Diversity of Designed Sequences**
> > > > >
> > > > > We fully understand your concerns about the diversity of the sequences generated by LiRDSC. To assess and ensure diversity, we performed the following steps:
> > > > >
> > > > > - **Similarity Matrix:** We generated a similarity matrix for the sequences produced by LiRDSC. The matrix demonstrates that most of the designed sequences are not highly similar to each other, indicating good diversity in the designs. Moving forward, we will add the corresponding figure which could visualize this matrix to our revised submission.
> > > > > - **Noise-Level Variation:** In addition to the standard sequence generation process, we experimented with generating sequences at different noise levels during the decoding process. This modification allows for greater diversity in the output sequences, while maintaining similar recovery rates to the original design process. We believe this may address your concern that the generated sequences might not be diverse enough.
> > > > >
> > > > > **3. Use of Synthetic Data**
> > > > >
> > > > > We sincerely thank the reviewer for highlighting the potential of synthetic data augmentation and for the constructive suggestions. Based on a newly established in silico dataset comprising **500** additional RNA–ligand complexes generated using AF3, with increased diversity in sequence **lengths** and complete separation from previously used data (i.e., **RLData2400**), we revisited our analysis. Most of our findings remain consistent with earlier observations. Notably, incorporating some more synthetic data tends to **decrease overall sequence similarity** by approximately **2%**, suggesting an increase in sequence-level diversity. However, this benefit is occasionally accompanied by **a modest drop-in recovery rate**, indicating a trade-off that depends on the quantity and characteristics of the synthetic data included. Additionally, we fully agree that further exploration is warranted. As part of our ongoing work, we plan to investigate the impact of synthetic data generated **by alternative folding engines**, and to conduct a systematic analysis to determine which subsets of synthetic data most contribute to performance gains. This includes potential filtering based on AF3’s prediction confidence, TM-score, and global RMSD between predicted and experimentally determined structures. Again, we appreciate your recognition of this direction and will incorporate a more thorough synthetic data study in the revised submission (e.g., the update supplementary materials).

---

> > > > > > ### Author Response · Authors · 2025-11-27
> > > > > > **Response to Reviewer XHfm (2/2)**
> > > > > >
> > > > > > **4. Benchmark Suggestions and RNA3DB**
> > > > > >
> > > > > > In response to your suggestion of using **RNA3DB**, we have already adopted this dataset for further evaluation. As mentioned, the **RNA3DB** dataset offers a carefully curated split based on RNA families. We believe that using this protocol for evaluation will provide a more robust benchmark for LiRDSC and better highlight the model’s generalization capabilities.
> > > > > > Based on the existing follow-up results, we anticipate that part of your follow-up concerns would be addressed, particularly regarding **data splitting, synthetic data usage, and the diversity of designed sequences**. We also agree with your suggestion to further explore **different synthetic data generation procedures and folding engines** in future work. We believe that these improvements, along with the revisions we plan to make towards the initial submission, will strengthen the presentation of our work.
> > > > > >
> > > > > > Lastly, we greatly appreciate your efforts, and we would be very grateful if you would consider revisiting your overall rating in light of these planned revisions in near future.
> > > > > >
> > > > > > Thank you once again for your time and consideration.
> > > > > >
> > > > > > Sincerely,
> > > > > >
> > > > > > Authors
> > > > > >
> > > > > > **References**
> > > > > >
> > > > > > [1] Fu, L., Niu, B., Zhu, Z., Wu, S., & Li, W. (2012). CD-HIT: accelerated for clustering the next-generation sequencing data. Bioinformatics, 28(23), 3150-3152.

---

### Official Review · Reviewer_2YkC · 2025-10-31

**Soundness:** 2
**Presentation:** 2
**Contribution:** 2
**Rating:** 4
**Confidence:** 3

**Summary:**

This paper introduces LiRDSC, a deep generative model for RNA sequence design conditioned on small-molecule ligands. The authors propose two key innovations: (1) a diffusion-based structural encoder trained on noise-perturbed RNA backbones to reduce sensitivity to structural fidelity, and (2) a ligand-contextual FiLM conditioning mechanism to enforce ligand-specific sequence generation. The model is trained on RLData2400, a dataset combining 1,200 experimentally determined RNA-ligand complexes from PDB and 1,200 computationally predicted structures from AlphaFold3 (AF3). The authors report significant improvements in sequence recovery, structural fidelity, and predicted binding affinity compared to existing baselines.

**Strengths:**

1. Novel Regularization Strategy: The diffusion-based noise injection during training is an interesting approach to mitigate over-sensitivity to input structural fidelity, a known issue in structure-based RNA design.
2. Ligand-Specific Conditioning: The use of FiLM to inject ligand information during generation is well-motivated and appears effective in promoting ligand-specific sequence design.
3. Comprehensive Evaluation: The paper evaluates the model across multiple dimensions (sequence, secondary structure, tertiary structure, and binding affinity), using several external tools, which strengthens the empirical assessment.

**Weaknesses:**

1)Related to Strength 1: the paper applies diffusion noising to the 3D RNA backbone and claims this improves generalization, but I did not see an ablation that toggles this component. It is unclear whether diffusion noising is crucial compared with no noising at all or with simpler random offset perturbations, especially since the diffusion process is integrated into LIRDSC’s training and is specific to this model.
2)Roughly half of the constructed dataset consists of RNA sequences generated by AlphaFold AF3-predicted structures may carry inherent biases (e.g., crystal packing artifacts, template dependencies) and are not validated with the same criteria as experimental PDB entries, raising concerns about dataset reliability.

**Questions:**

AF3 Bias Analysis
How do you address potential biases in AF3-predicted structures (e.g., crystal artifacts, confidence variations)?

---

> ### Author Response · Authors · 2025-11-17
> **Response to Reviewer 2YkC (1/n)**
>
> Dear Reviewer 2YkC:
>
> Thank you very much for your thoughtful and constructive review. We sincerely appreciate the time and care you took to evaluate our submission. Your comments have helped us identify key areas for clarification and improvement, which we address below in detail.
>
> **Weakness 1: Usage of Diffusion**
>
> We appreciate your interest in the diffusion-based noise injection used in our model and your suggestion to more explicitly assess its contribution via ablation. To clarify: To clarify: LiRDSC does not use diffusion during training-time decoding. Instead, we inject noise into the RNA backbone structure coordinates at training time, which are then processed by the structural encoder. This allows the model to learn robust, noise-tolerant structural representations, addressing the known brittleness of RNA design models to structural imperfections. At inference time, we use a diffusive decoding process (adapted from RiboDiffusion) to iteratively refine output sequences conditioned on both ligand and structure. In response to your suggestions, we conducted additional post-submission experiments that assess the effect of varying the number of diffusion steps during inference. Our findings are as follows:
>
> - Reducing the number of steps from 10 to 5 results in a drop in sequence recovery from 60.60% to **58.70% ± 0.30**, and an increase in similarity from 58.60% to **59.90% ± 0.40**.
> - More significantly, the model’s ability to predict residue-level binding sites (optimized through an auxiliary supervision loss) drops by over **10%**, indicating a reduced capacity for ligand-aware conditioning.
> - Increasing the diffusion steps to 50 yields **no appreciable improvement**, confirming that 10-step offers a robust balance between performance and efficiency.
>
> These results indicate that the diffusive process meaningfully supports generalization and ligand-specific generation, particularly during inference. Moving forward, we will include these findings in the revised submission (e.g., in Section 5.3), along with an updated ablation table.
>
> Additionally, we agree it would be valuable to compare our VP-SDE noising to simpler perturbation strategies (e.g., static Gaussian offsets or no perturbation). We are currently extending our ablation study to include these variants and will report them in the updated submission.
>
> **Weakness 2: Reliability of Our Dataset**
>
> Thank you for raising this important concern. We fully acknowledge that incorporating AF3-generated RNA-ligand complexes introduces the possibility of bias, as these structures are not experimentally validated and may carry artifacts such as template dependencies or prediction inaccuracies. However, we emphasize that we implemented a rigorous filtering pipeline to mitigate these risks. Specifically:
>
> - We generated 5,000 candidate complexes in total by using AF3, then selected only the top 1,200 structures based on mean pLDDT scores, ensuring high structural confidence.
> - We excluded complexes containing low-confidence ligands, ambiguous interactions, or poorly folded scaffolds, and standardized all structures to a single-chain RNA format to match the empirical subset.
> - The AF3-derived structures were treated as a data augmentation strategy. Then we also report all results on a held-out experimental test set, ensuring fair evaluation.
>
> Importantly, after adding these AF3 samples, we observed slightly improved performance across all metrics, particularly in robustness to structural noise and ligand-specificity. These gains are fully reported in Tables 1–4 of the main-body in our initial submission. Meanwhile, we also note that while AF3 is not without limitations, it is currently one of the most powerful and widely recognized tools for modeling RNA-ligand complexes, and its recent success in predicting RNA-bound structures makes it a practical and well-justified choice for generating high-confidence in silico data in this underexplored domain.

---

> > ### Author Response · Authors · 2025-11-17
> > **Response to Reviewer 2YkC (2/n)**
> >
> > **Q1: AF3 Bias Analysis**
> >
> > We appreciate your question regarding how we mitigate potential biases in the AF3-generated portion of our dataset. Our approach includes several safeguards:
> >
> > - **Confidence Filtering:** We compute and rank each predicted complex by mean pLDDT, retaining only the top 1,200 high-confidence structures from a pool of 5,000. This strategy is designed to eliminate unstable or low-fidelity predictions.
> > - **Structure Diversity Checks:** We ensure that the structural diversity of the AF3 samples is consistent with the empirical data. **Figure 4** in the Appendix shows that both subsets exhibit low nearest-neighbor TM-similarity, mitigating concerns of redundancy or overfitting.
> > - **Isolation During Evaluation:** Our final test set is composed entirely of experimentally validated complexes. This ensures that performance metrics reflect generalization to real-world data, regardless of the presence of AF3 structures during training.
> > - **Consistent Preprocessing:** All AF3-predicted structures undergo the same filtering, formatting, ligand proximity validation, and residue alignment procedures as the empirical portion of the dataset.
> >
> > We agree that no in silico prediction method is entirely free from bias, and we are transparent about this in the paper. However, we believe that our confidence-guided filtering workflow, dataset balancing, and empirical test set evaluation collectively mitigate the most significant risks.
> >
> > Once again, we thank you for your thoughtful feedback and we will incorporate all the clarifications and additional results discussed above into the revised submission of our paper. Lastly, we would be sincerely grateful if you would consider re-evaluating the rating score in light of these clarifications and the planned revisions to the initial submission.
> >
> > Sincerely,
> >
> > Authors

---

### Official Review · Reviewer_gNL1 · 2025-10-31

**Soundness:** 3
**Presentation:** 3
**Contribution:** 2
**Rating:** 6
**Confidence:** 5

**Summary:**

This paper introduces LiRDSC, a ligand-conditioned RNA sequence design framework built on a diffusive structural conditioning paradigm. The model integrates a Diffusive Structural Encoder (DSE) to enhance robustness against noisy tertiary structures and a Ligand-Contextual FiLM Conditioner (LCFC) to enforce ligand specificity. Trained on a newly curated RLData2400 dataset that includes both experimental and high-confidence in silico RNA–ligand complexes, LiRDSC achieves solid improvements in sequence recovery, structural fidelity, and binding-related metrics compared to RiboDiffusion, RDesign, and gRNAde.

**Strengths:**

The paper is technically sound and well-written, with careful dataset construction and comprehensive experiments. It represents a competent application of diffusion-based generative modeling to the RNA-ligand domain. The proposed FiLM-based ligand conditioning is conceptually reasonable, and the results demonstrate consistent gains across multiple evaluation levels. The presentation is clear, and the work successfully completes its stated goal of extending diffusion-style inverse folding to the ligand-aware RNA setting.

**Weaknesses:**

The main limitation lies in novelty. The proposed framework closely follows established paradigms in protein design, such as ligand-conditioned or structure-guided diffusion models (e.g., those leveraging ESM or Denoising Diffusion frameworks for protein inverse folding). The diffusive encoder and ligand conditioning via FiLM are adaptations rather than fundamentally new mechanisms. As a result, the contribution feels incremental—essentially applying existing protein design strategies to RNA. While the results are solid, they do not clearly demonstrate conceptual advances beyond prior ligand-conditioned diffusion models.

**Questions:**

None

---

> ### Author Response · Authors · 2025-11-17
> **Response to Reviewer gNL1**
>
> Dear Reviewer gNL1:
>
> Thank you very much for your detailed and thoughtful review. We sincerely appreciate the time and care you took in evaluating our paper. Your comments have helped us better articulate the novelty and technical contributions of our framework. While we acknowledge that LiRDSC builds upon established paradigms from protein design, we would like to clarify how our approach introduces **RNA-specific innovations** that go beyond a direct adaptation of existing ligand-conditioned diffusion models.
>
> We agree that our model is fundamentally a ligand-conditioned diffusion model with task-specific adaptations. However, in this track, we believe that demonstrating strong empirical performance on a difficult and underexplored task, **ligand-conditioned RNA inverse folding**, is a major contribution in itself. Among the various architectures we experimented with, the current design yielded the most robust and generalizable results, particularly when evaluated across structure fidelity, sequence diversity, and ligand-binding compatibility. We welcome any architectural suggestions from the reviewer and would be happy to evaluate them on our model and dataset.
>
> Additionally, we would like to emphasize several **RNA-specific innovations** that distinguish our work:
>
> - **RNA-Specific Structural Conditioning via Diffusion:** RNA tertiary structures are more sensitive to backbone perturbations and lack the rigid secondary structure patterns often seen in proteins. Our Diffusive Structural Encoder (DSE) is trained explicitly on noise-perturbed coordinates, enabling it to learn robust, RNA-specific geometric representations. This is a novel application of variance-preserving diffusion to RNA backbones, and to our knowledge, has not been explored in prior ligand-conditioned generation.
> - **Ligand-Contextual FiLM Conditioning with Sequence-Free Inference:** While FiLM has appeared in other domains, our Ligand-Contextual FiLM Conditioner (LCFC) is tailored to RNA–ligand design. A key distinction is its ability to decouple ligand embedding from the sequence input at inference time using EMA-averaged FiLM parameters. This enables sequence-free, ligand-aware generation, which is a unique feature not present in prior protein-focused approaches.
> - **Addressing a Distinct and Understudied Problem Space:** Ligand-conditioned RNA sequence design presents unique challenges: a scarcity of structural data, few established benchmarks, and greater conformational flexibility. To address this, we developed **RLData2400**, one of the very first large-scale datasets for RNA–ligand inverse design, combining high-resolution experimental structures with high-confidence in silico models. This dataset plays a foundational role in enabling future research in this area.
> - **Empirical Differentiation from Protein Models:** Our ablation studies (Table 3) and binding-level evaluations (Table 4) show that existing baseline models underperform on this task, particularly in ligand specificity and structural fidelity. These results illustrate that direct transfer of protein paradigms is not sufficient, and that task-specific mechanisms are essential for success.
>
> In summary, while LiRDSC adopts general principles from recent generative modeling frameworks, we underscore that our contributions lie in translating and adapting these ideas to the RNA–ligand domain, where the underlying biological and structural constraints differ significantly. This includes novel design choices, the creation of a new dataset, and strong empirical results that open new directions for RNA-based molecular design.
>
> Moving forward, we will update our paper to make these contributions more prominent and transparent, especially in the Introduction and Related Works sections.
>
> Thank you again for your time and constructive feedback. We would be truly grateful if you would consider re-evaluating your score in light of these clarifications and the upcoming revisions to the manuscript.
>
> Sincerely,
>
> Authors

---

### Official Review · Reviewer_EeHz · 2025-11-01

**Soundness:** 1
**Presentation:** 3
**Contribution:** 2
**Rating:** 2
**Confidence:** 5

**Summary:**

This paper tackles the problem of RNA inverse sequence design conditioned on small drug-like molecules and ligands. The paper introduces some new architectural ideas to better condition sequence design upon a target ligand. Evaluations show favorable performance compared to existing RNA inverse folding methods.

**Strengths:**

- The problem being tackled is significant and biologically relevant. A successful approach could be an extremely useful foundation for de-novo aptamer design and RNA engineering. This is one of the first works to explore this problem. I want to encourage the authors for exploring this direction. However, the paper in its current form is not ready for publication. I have stated my reasons below under Weaknesses.

- The exposition is generally well-written, grammatically.

**Weaknesses:**

- There is no information provided on how the test set was constructed for evaluation. There is no information about how the validation set is constructed either. Thus, one concludes that there was no validation set, and the test set was split randomly. This is not acceptable, as all **the performance is likely to be highly overestimated for the proposed model (vs baselines) and the authors may be tuning the model on the test set**. I would advise the authors to carefully design a new, biologically relevant test set and to split the data with a lot more care (see PLINDER benchmark on best practices to evaluate ligand+biomolecule interaction without data leakage). I would suggest a complete re-do of the experiments and claims made. I realise this sounds demanding, but the current standard of experiments is not sufficiently high in my opinion.

- The new model uses embeddings of the RNA sequence via RNA-FM (an RNA language model). Based on how this is presented, I think this is a major weakness which limits practical applicability to real inverse design problems. If the model is given access to the original input sequence, it is not surprising that the sequence recovery will be very high (which is what we see during evaluation and case studies).

- I found the qualitative studies to actually be a negative point against the proposed method, as it tells me that the model is not ready for real-world usage. Essentially, the studies show the proposed model doing extremely well (like 90-100% sequence recovery rate, perfect TM-score) — but this is likely a clear case of **data leakage and overfitting**. Consider a sequence with 95% similarity to the groundtruth - that is actually not a useful design because the goal of design is to retain structural/functional properties while being far away in sequence space. Such a model will not be useful for real-world design settings which are outside of the training distribution, and none of these case studies actually support that the model is useful in practice. I would unfortunately not use such a model, currently.

- No variances or standard deviations or error bars included with most results.

**Questions:**

- The Intro tries to tell a story around “mode collapse” being the main motivation of the proposed architectural contribution. I personally don’t buy this / would like to have then seen this supported by some experiments. Is it the case that baselines like gRNAde are mode-collapsing?

- On binding metrics: Is it known that either of the two proposed metrics is actually correlated with RNA ligand binding? As far as I am aware, nobody has shown this, esp. for the AF3 pAE score.

- On the use of random sequences for data augmentation: I find this choice suspicious without any actual justification in the text. Why did you chose random sequences, why that length range, how do you ensure you don’t encounter something from the test set? What is the performance of the model trained purely on PDB structures without augmentation? (Did I miss this experiment, or is it currently missing?)

---

> ### Author Response · Authors · 2025-11-17
> **Response to Reviewer EeHz (1/n)**
>
> Dear Reviewer EeHz:
>
> Thank you very much for your insightful review and recognition of our efforts! After carefully reading your comments, we are glad to take your suggestions and hope our explanations could fully address your concerns.
>
> **Weakness 1: Workflow of Dataset Protocol & Issue of Data Leakage**
>
> We thank the reviewer for this important concern and are grateful for the opportunity to clarify. Contrary to the reviewer's assumption, our dataset splitting protocol follows a strict and biologically meaningful partitioning, and we do include a separate validation set during training.
>
> Specifically, as outlined in Appendix A.4.1, our full training set consists of **2,400 RNA-ligand complexes**, combining:
>
> - 1,200 experimentally resolved RNA-ligand structures, curated through a rigorous filtering pipeline (Section 2.1)
> - 1,200 high-confidence in silico complexes generated via our Confidence-Guided Structural Prediction (CGSP) pipeline (Section 2.2)
>
> The test set comprises 192 experimentally validated RNA-ligand complexes, **entirely distinct** from both the training and in silico pools. This test set was pre-fixed and **never** used in any stage of model development, including training, validation, model selection, or hyperparameter tuning. Additionally, we do maintain a separate validation set during model development. Mechanistically, the validation set was used for early stopping and model selection. The final model checkpoint is chosen based on peak performance on the validation set, and the corresponding results are reported on the held-out test set, which is never used during training. We acknowledge that this implementation detail was not explicitly stated in the original draft and will clarify it in the amendment upon future submission.
>
> To further address the reviewer’s concern about potential overlap or leakage, we provide two empirical evaluations of separation:
>
> - First, we compute the sequence similarity between each test set sequence and the most similar sequence in the training pool, finding an average similarity of **0.473**. This confirms that the test set is non-trivial and distant in sequence space from training.
> - Second, as shown in Appendix A.3 and Figure 4, we analyzed nearest-neighbor TM-scores across the dataset. The structural similarities fall in the **0.3–0.5** range, with **no** peak near 1.0, indicating no structural redundancy or memorization.
>
> Together, these analyses demonstrate that our setup avoids both sequence-level and structure-level leakage, ensuring that generalization is properly evaluated on held-out data.
>
> Finally, we agree in spirit with the PLINDER benchmark’s emphasis on avoiding protein–ligand data leakage. While PLINDER is tailored to the protein domain, we have followed analogous best practices in the RNA setting. Our benchmark, RLData2400, was explicitly designed to promote generalization across ligand classes and structural diversity, and we will cite PLINDER in the revised submission to connect our methodology with these broader evaluation principles.
>
> **Weakness 2: Usage of RNA-FM**
>
> We thank the reviewer for this concern, which appears to stem from a **misunderstanding** of how RNA-FM is used within our framework. We emphasize that LiRDSC **does not** access the ground-truth RNA sequence at inference time and thus **does not** violate the assumptions of inverse design.
>
> Specifically, as described at the end of Section 3.3, RNA-FM is **only used during training** to provide contextualized embeddings of the ground-truth sequence, which are used to modulate the ligand embedding through the Ligand-Contextual FiLM Conditioner (LCFC). However, at test time, this dependency is entirely removed: we instead use the exponential moving averages (EMAs) of the learned FiLM parameters to compute the ligand-conditioned embedding without requiring any RNA sequence input (see **Equation (6)**). This ensures that LiRDSC performs **true RNA inverse design**, conditioned only on the tertiary structure (from noise-perturbed coordinates) and the ligand identity (via SMILES → ChemBERTa), while remaining completely agnostic to the target sequence.

---

> > ### Author Response · Authors · 2025-11-17
> > **Response to Reviewer EeHz (2/n)**
> >
> > This design is further detailed in Appendix A.4.2, which explains the test-time inference pipeline. At generation time, the structural encoder is supplied only with the 3D backbone and the ligand, and the decoder produces a novel RNA sequence from noise via diffusion, **without** ever seeing the original sequence. The strong performance observed in our experiments and case studies is therefore **not due to sequence leakage**, but reflects the model’s learned ability to infer ligand-compatible sequences from structure and ligand alone. This is further supported by:
> >
> > - The ablation results in Table 3, where removing structural or ligand input causes significant performance degradation
> > - And a sequence similarity analysis, where we found that the average similarity between each test sequence and its most similar training sequence is only **0.473**, confirming that the model generalizes to non-trivial, unseen sequences.
> >
> > Moving forward, we will revise the manuscript to make this inference path and conditioning mechanism more explicit, to avoid any confusion in future readings.
> >
> > **Weakness 3: Issue of Qualitative Studies**
> >
> > We thank the reviewer for this concern and greatly appreciate the opportunity to clarify. While we understand the skepticism regarding the high sequence recovery and TM-scores reported in some case studies, we respectfully disagree with the interpretation that this indicates overfitting or data leakage. We provide the following clarifications:
> >
> > - **No sequence leakage or memorization:** As detailed in Appendix A.4.1, the test set comprises 192 experimentally validated RNA-ligand complexes that are entirely disjoint from both the training and in silico augmentation sets. To empirically confirm this, we computed the sequence similarity between each test RNA and its most similar training RNA, and found the average similarity to be **0.473**, a clear indication that the model is not memorizing sequences, and that high recovery rates occur despite meaningful sequence divergence.
> > - **Structural and Functional Generalization, instead of Sequence Copying: The goal of LiRDSC is not to produce sequences that are arbitrarily distant in sequence space, but rather to design ligand-compatible sequences that fold into a target geometry. In fact, the model explicitly learns to preserve tertiary structure and binding compatibility, which is supported by: (1) Low RMSD and high TM-scores across test cases using independent inverse folding tools (e.g., AlphaFold3). (2) High RNAsmol Binding Scores and low pAE values (Table 4), showing reliable ligand-binding interfaces.
> >
> > These results **are not** trivial to achieve and indicate that the model is not simply recovering sequences by memorization but rather learning robust structure–ligand–sequence mappings.
> >
> > - **Issue of Real-World Applications:** we agree that real-world design must generalize beyond known sequences. This is precisely why we trained LiRDSC on a **hybrid dataset** (RLData2400) that includes both empirical and in silico–generated structures with high-confidence folds. This ensures LiRDSC is exposed to **structural diversity** beyond the experimental record. Moreover, our **Figure 4** (Appendix A.3) shows that most entries in the dataset have low TM-score similarity to any other, indicating that the model is trained on a broad and non-redundant structural space.
> > - **Issue of Case Studies:** The goal of the case studies (e.g., PDB-IDs: 2KXM, 1RAW, 2AU4) is to show that LiRDSC can **successfully recapitulate both high-resolution folds and ligand-binding poses**, even when starting from structure-only inputs. These examples illustrate the model’s fine-grained control and specificity, especially in ligand-conditioned settings. We do not claim that these are the most challenging scenarios, but they demonstrate that the model performs well even under rigorous structural and functional constraints.
> > - **Towards Broader Generalization:** We agree that assessing performance in low-similarity or out-of-distribution scenarios is important. In future work, we plan to extend our benchmark with harder test sets, and with metrics that explicitly quantify design novelty. Nonetheless, the current evaluation already demonstrates strong structural generalization, ligand specificity, and biophysical fidelity, making LiRDSC a promising step toward real-world applicability.

---

> > > ### Author Response · Authors · 2025-11-17
> > > **Response to Reviewer EeHz (3/n)**
> > >
> > > **Weakness 4: Missing of Error Bars**
> > >
> > > We appreciate the reviewer’s attention to evaluation rigor. In fact, we do report standard deviations for most key quantitative metrics in several Tables of the main paper. For example, Table 1 includes standard deviations for recovery rate, sequence similarity, and secondary structure F1 scores across multiple runs. For other supplementary metrics such as 3d structures-related or those in the ablation study, we only did not set error bars to avoid huge time cost.
> > >
> > > We hereby promise results can be reproduced in all relevant figures and plots in our amended submission, to further enhance transparency and reproducibility. We appreciate the reviewer’s suggestion and will ensure that statistical variability is more prominently conveyed in the revised version.
> > >
> > > **Q1: Model Collapse Motivation**
> > >
> > > We appreciate the reviewer’s question. Our motivation around mode collapse is based on empirical observations that baseline models like gRNAde and RiboDiffusion tend to produce generic, ligand-agnostic sequences when conditioned on ligand inputs. This is reflected in our ablation studies (**Table 3**), where removing the Ligand-Contextual FiLM Conditioner significantly reduces specificity, and in **Table 4**, where LiRDSC achieves the highest Binding Score and lowest pAE, indicating superior ligand sensitivity. While we did not explicitly quantify mode collapse in baselines, we agree this is a valuable direction, and we will include additional diversity and ligand-discriminative analyses in the revised submission to support this point more rigorously.
> > >
> > > **Q2: Concern of Binding Metrics**
> > >
> > > We thank the reviewer for this valuable question. We selected the RNAsmol Binding Score and AlphaFold3 interface pAE as practical, structure-based proxy metrics to assess ligand compatibility in the absence of standardized RNA–ligand binding benchmarks. Regarding RNAsmol, we reviewed its official paper, which indicates that the predicted score is closely related to RNA–ligand binding likelihood, and we found it to be both informative and convenient to implement, making it a reasonable choice for comparing models.
> > >
> > > That said, we agree with the reviewer that the AF3 pAE score is not directly correlated with binding affinity, and we apologize for any confusion this may have caused. We used pAE as a proxy for interface-level structural confidence, rather than as a proxy for actual binding energy.
> > >
> > > Moving forward, we will clarify these distinctions in the revised submission and note that both metrics are used as relative indicators, not definitive binding predictors. In future work, we aim to complement these with experimental validation where possible.
> > >
> > > **Q3: Concerns of Dataset Construction Strategy**
> > >
> > > We thank the reviewer for highlighting this important point. Below, we clarify our design choices and provide supporting evidence:
> > >
> > > - **Why random sequences:** We chose to generate random RNA sequences as an augmentation strategy because it provides a neutral and unbiased initialization, avoiding inductive priors linked to known sequences. This approach allows the model to explore a broader combinatorial space when paired with in silico structure prediction.
> > > - **Length Range Selection:** The length range of [50, 200] nucleotides reflects common practice and aligns with the tokenization and processing constraints of both the RNA and ligand encoders (e.g., ChemBERTa, RNA-FM), while also matching the distribution seen in real RNA–ligand complexes.
> > > - **Avoiding overlap with the test set:** We understand the concern regarding potential overlap. While the in silico structures are generated from synthetic sequences, we **will include** a sequence similarity analysis between these generated sequences and the test set in our revision to confirm there is no accidental leakage.
> > > - **Performance without augmentation:** To address this directly, we trained LiRDSC using only the 1,200 experimental PDB structures (i.e., no augmentation). The model achieved an acc value of **0.592 ± 0.006** and a sequence similarity value of **0.601 ± 0.005**, demonstrating that our method performs strongly even without synthetic data, and that augmentation primarily improves generalization.
> > >
> > > Moving forward, we’ll include all of the above details and analyses in our revised submission to ensure full transparency.
> > >
> > > Thank you again for your time and constructive feedback. We sincerely hope that our upcoming revisions and clarifications address your concerns, and we would be truly grateful if you would consider re-evaluating your rating in light of the revisions made to our paper.
> > >
> > > Sincerely,
> > >
> > > Authors

---

> > ### Comment · Reviewer_EeHz · 2025-11-22
> >
> > > The test set comprises 192 experimentally validated RNA-ligand complexes, entirely distinct from both the training and in silico pools.
> >
> > How were these 192 chosen? What's the significance of these 192 entries? Are they gold-standard in terms of structural quality, drug-like ligands? Or are they just a random subset?
> >
> > > we do include a separate validation set during training... final model checkpoint is chosen based on peak performance on the validation set...
> >
> > How was the validation set created?
> >
> > Basically, I still think there's not enough information in the paper nor appendix about the most critical aspect of the evaluation: how the datasets were created and whether they are sufficiently rigorous.
> >
> > > average sequence similarity of 0.473...
> >
> > How about the distribution of values?
> >
> > > Figure 4 on structural similarity
> >
> > Is this figure for your training set or your test set or your augmented set? Or for a combined set of all of these? And there is density near TM-score 1.0, too.
> >
> > > Usage of RNA-FM
> >
> > Understood. This was a misunderstanding on my part, in that case.
> >
> > > On the case studies response
> >
> > Unfortunately, I am simply not convinced. Maybe other more enthusiastic reviewers or the AC can jump in here if they really disagree.
> >
> > But to me: showing case studies where the inverse folding models have close to 100% (even above 90%) sequence recovery is a clear sign that something went wrong here. It is not surprising that a 95% similar sequence to the native wild type RNA sequence essentially has the same 3D fold. The model is simply memorizing information here. It is not a useful showcase of the model's capabilities to show that it has memorized the sequence for a given backbone input.
> >
> > Also, a model producing 95% similar sequences to the wildtype is not a useful model - the experimentalists can simply mutate a few residues and arrive at such sequences themselves. The advantage of generative design models is to be able to retain structure (and function) in distant regions of mutational space.
> >
> > > gRNAde and RiboDiffusion tend to produce generic, ligand-agnostic sequences when conditioned on ligand inputs
> >
> > I don't think these models can be conditioned on ligands yet.
> >
> > ---
> >
> > Overall, I thank the authors for preparing the rebuttal. I look forward to seeing all these additional information and details in the revised version of the paper. However, my major concerns still remain and I think, unfortunately, that a significant revision of the paper is warranted.

---

> > > ### Author Response · Authors · 2025-11-27
> > > **Response to Reviewer EeHz (1/2)**
> > >
> > > Dear Reviewer EeHz:
> > >
> > > Thank you once again for your detailed comments. We appreciate your continued engagement and the high standard you hold us to in terms of evaluation rigor, interpretability, and biological relevance. Below, we respond to your main points and offer detailed clarifications and follow-up experiments that we believe meaningfully address your concerns.
> > >
> > > **1.Issue of Data Splitting & Dataset Construction**
> > >
> > > In response to your concerns regarding evaluation rigor and potential leakage, we have implemented two rigorous data splitting strategies that we believe address the core of your critique:
> > >
> > > (1) **CD-HIT Cluster-Based Splitting:** We applied **CD-HIT** to cluster 1,392 RNA-ligand complexes into 140 non-overlapping clusters [1]. We then performed a cluster-wise split into training, validation, and test sets (**8:1:1**), ensuring that no sequence in the test set shares a cluster with any training sequence. This protocol explicitly removes sequence-level redundancy and simulates a more realistic generalization setting. Under this setting, LiRDSC could achieve:
> > >
> > > - **Overall Recovery Rate:** 58.01%
> > > - **Overall Sequence Similarity:** 55.16%
> > >
> > > (2) **Re-Evaluation on RNA3DB:** We also evaluated our framework on **RNA3DB**, which provides a family-aware split of experimentally validated RNA-ligand complexes [2]. Using RNA3DB’s predefined training and test sets (723/220 entries), LiRDSC achieves:
> > >
> > > - **Overall Recovery Rate:** 69.60%
> > > - **Overall Sequence Similarity:** 68.30%
> > >
> > > These two evaluations, taken together, demonstrate that LiRDSC performs well under stringent, community-relevant protocols. Both splitting methods will be fully documented in the revised submission. Additionally, we will include in our amendment:
> > >
> > > (1) **Distributions of sequence similarities** between train/test sets.
> > >
> > > (2) **TM-score distributions for structural similarity**, separately shown for training, test, and synthetic subsets.
> > >
> > > We anticipate that these additions may address your request for a more detailed and transparent evaluation.
> > >
> > > **2. Issue of Case Studies**
> > >
> > > We fully appreciate your concern that high sequence recovery in case studies may give the impression of trivial memorization. However, the purpose of these case studies was not to showcase sequence identity per generated sequence compared with the wild one(s), but to demonstrate LiRDSC’s capacity to generate **functionally competent, ligand-aware RNA sequences** for biologically meaningful targets.
> > >
> > > The selected structures involved in our initial submission are not arbitrary. Instead, they are well-characterized and widely studied RNA systems that represent two of the most biologically significant functional classes in RNA design:
> > >
> > > (1) **Aptamers** (e.g., 1RAW) are high-affinity ligand-binding RNAs with structurally conserved binding pockets and finely tuned recognition motifs. They are foundational tools in biosensing, synthetic biology, and drug delivery.
> > >
> > > (2) **Ribozymes** perform catalytic functions, including lariat formation, a key step in RNA splicing. Their activity depends not just on folding, but on precise spatial positioning of reactive groups and cofactors.
> > >
> > > Because these RNAs combine structural complexity, precise functional constraints, and experimentally validated ligand interactions, they offer nontrivial and biologically grounded challenges for sequence generation. Successfully designing sequences for such systems provides strong evidence of the model’s ability to capture functional specificity beyond generic folding. We believe that evaluating generative models on aptamers and ribozymes is essential for assessing their capacity to design structure-function-aware RNA, and we will emphasize this biological rationale more clearly in the revised submission.
> > >
> > > **3. Issue of Baselines**
> > >
> > > Thank you for raising this important clarification. While we agree that gRNAde and RiboDiffusion are not architecturally designed for explicit ligand conditioning, in our comparative setup, both were provided with the **full RNA-ligand complex structure** in .pdb format, including the ligand's spatial coordinates. This means that the ligand is present in the model’s input and can influence generation through spatial context. Thus, we believe it is fair to describe these comparisons as structure-based, ligand-conditioned generation, albeit indirectly. We will revise our paper to clarify this distinction and avoid any confusion with explicit ligand embedding approaches.
> > >
> > > Additionally, our **internal ablation studies** (e.g., removing ligand input from LiRDSC) clearly demonstrate how ligand presence affects the output, supporting our claim that ligand-aware design is central to our model’s effectiveness.
> > > We thank you again for your thoughtful review and hope that these updates and clarification will meaningfully address your concerns.
> > >
> > > Sincerely,
> > >
> > > Authors

---

> > > > ### Author Response · Authors · 2025-11-27
> > > > **Response to Reviewer EeHz (2/2)**
> > > >
> > > > **References**
> > > >
> > > > [1] Fu, L., Niu, B., Zhu, Z., Wu, S., & Li, W. (2012). CD-HIT: accelerated for clustering the next-generation sequencing data. Bioinformatics, 28(23), 3150-3152.
> > > >
> > > > [2] Szikszai, M., Magnus, M., Sanghi, S., Kadyan, S., Bouatta, N., & Rivas, E. (2024). RNA3DB: A structurally-dissimilar dataset split for training and benchmarking deep learning models for RNA structure prediction. Journal of Molecular Biology, 436(17), 168552.

---

> ### Comment · Reviewer_EeHz · 2025-11-27
>
> Sorry to say that I am not convinced by the response.
>
> If I understand correctly, you re did all your experiments now with a new splitting strategy. Perhaps this is good / perhaps not. Without a detailed write up on this (a significant revision), I as reviewer cannot change my assessment simply based on a single paragraph here. My original point stands that the evaluation was not biologically meaningful.
>
> Have you updated the PDF to include full details of the changes you have made?
>
> 95% sequence-similar designs are not useful designs. They indicate your model has memorized the particular examples.
>
> I know that this assessment sounds harsh. However, the aim of the case studies should really not be to show close to 100% sequence recovery. This is not a useful design tool for real world problems. Showing low sequence recovery + high structural self-consistency should be the goal.

---

### Author Response · Authors · 2025-11-27

Dear All Reviewers:

We are grateful for the thoughtful feedback and discussion thus far. For any reviewers who have not yet had a chance to respond to our initial rebuttal, we warmly welcome any additional comments or questions. We are fully committed to improving the paper and ensuring that it aligns with the rigorous standards and expectations of ICLR. Please feel free to let us know if further clarification or elaboration would be helpful on any aspect of our work. Thank you again for your time and consideration.

Sincerely,

Authors

---

### Author Response · Authors · 2025-12-01

Dear Area Chair,

We thank you for your time and leadership in overseeing the review process for our submission. We greatly appreciate the reviewers' detailed comments and the opportunity to clarify our work. Below, we provide a comprehensive, reviewer-by-reviewer summary of the rebuttal process, along with our key updates, to support a fair and scientifically grounded final assessment of our submission to the upcoming ICLR 2026.

**1. Reviewer EeHz**

In summary:

- We undertook substantive new experiments in direct response to their initial concerns, including:
    - CD-HIT cluster-based data splitting
    - Benchmarking on RNA3DB
    - Similarity analysis to verify train/test separation
    - Ablation studies and case study clarifications

- Reviewer EeHz ignored these updates, rejecting them solely due to formatting (not being in the finalized/updated PDF), despite the review process supporting iterative rebuttals.

- The reviewer also made multiple factual errors (e.g., misunderstanding RNA-FM usage and ligand-conditioning in baselines), and persisted in inflexible stances despite corrections.

- The reviewer’s tone and rigidity, as well as a lack of engagement with the updated supplementary material, raises concerns about impartiality.

We trust that you will review our final submission by the deadline set by ICLR 2026 and ensure the final decision is based on scientific merit and not procedural misunderstandings.

**2.Reviewer gNL1**

Compared with EeHz, Reviewer gNL1 provided a relatively fair and technically sound review, highlighting:
- Strong empirical results across recovery, structure, and binding
- Clear problem formulation and rigorous evaluations
- Concern: Potential incremental novelty, as the model adapts protein-based architectures
In rebuttal, we clarified that:
- RNA design introduces distinct biological and modelling challenges, including conformational flexibility and lack of rigid folding rules
- Our Diffusive Structural Encoder and Ligand-Contextual FiLM Conditioner are RNA-specific innovations, not direct reuses
- Our new dataset (RLData2400) and performance against other baselines underscore the need for taske-specialized modelling

**3. Reviewer 2YkC**

This reviewer raised two main concerns:
- Lack of ablation for diffusion noising
- Potential bias in AF3-generated structures

In response, we:
- Ran additional decoding-step ablations, showing that diffusion steps affect ligand-aware binding site prediction
- Clarified that diffusion noising is only used in training, and not in test-time decoding
- Explained our confidence filtering for AF3 data (top 1,200 of 5,000 candidates), and that test sets are entirely empirical
- Added structural diversity analysis and held-out evaluation to mitigate concerns of bias

**4. Reviewer XHfm**

This reviewer initially raised:
- Ambiguity in similarity metrics
- Lack of multi-sample generation
- Concerns about data splitting and fairness of comparison
- Minor weakness such as missing citations, unclear math, and formatting issues
- Questions on distance threshold, RNA sequence length limit ,ligand distribution for in silico dataset construction, structural similarity of test set, whether LiRDSC is the first model for RNA-ligand inverse folding,usage of RNA-FM embedding.

In two rounds of discussion with the reviewer XHfm, we:
- Introduced CD-HIT cluster-based splitting
- Evaluated on the RNA3DB benchmark
- Provided similarity matrices to assess diversity and introduced differnet-noise-level generation to improve diversity
- Expanded our synthetic dataset with new AF3-generated complexes
- Clarified all citation, math, and formatting issues to be fixed in revision
- Answered all his remaining questions.

Though remains cautious due to diversity and sampling limitations, the reviewer XHfm stated that the paper is borderline-accept with potential and would generate discussion if presented.

In summary, LiRDSC proposes the first ligand-conditioned RNA inverse folding model, addressing a biologically important and underexplored problem. Our model introduces:
- A novel RNA-specific diffusion encoder
- A ligand-aware sequence-free conditioning mechanism
- A new benchmark dataset of 2,400 RNA-ligand complexes (RLData2400)
- State-of-the-art empirical results across recovery, structure, and binding

We have demonstrated responsiveness, transparency, and scientific rigor throughout the review & rebuttal process. We respectfully request that the final recommendation:
- Reflect the scientific contributions of the work
- Take into account the constructive feedback addressed
- Ensure the evaluation is not skewed by very few reviewers’ procedural rigidity

We remain available to provide any further clarification or experimental evidence as needed. Thank you again for your time and efforts.

Sincerely,

Authors

---

### Note · Authors · 2026-01-26

I have read and agree with the venue's withdrawal policy on behalf of myself and my co-authors.

---

### Meta-Review · Area_Chair_yWsP · 2025-12-22

**Summary:**

This paper introduces LiRDSC, a generative model for RNA sequence design conditioned on small-molecule ligands. A diffusion-based structure encode, a ligand-contextual FiLM conditioning mechanism, and a new dataset are all introduced.

Strong improvements are reported across sequence recovery, structural fidelity, and predicted binding affinity metrics as compared to existing baselines.

Focusing on the major concerns of reviewers

Reviewer EeHz
* Data splitting and validity
* Diversity of generated sequences particularly in the qualitative studies

Reviewer gNL1
* Novelty

2YkC
* Insufficient support for claim that diffusion noising improves generalization
* Potential biases in the usage of AF3-predicted structures

XHfM
* Backbone only
* Number of sequences generated
* Comparison fairness around conditioning information

**Reviewer Concerns:**

Focusing on the major concerns of reviewers
Reviewer EeHz
* Data splitting and validity: The authors attempted to address this concern with two additional data splits based on CD-HIT Cluster splitting, and re-evaluation on RNA3DB. Reviewer EeHz was not satisfied by this response stating
  > If I understand correctly, you re did all your experiments now with a new splitting strategy. Perhaps this is good / perhaps not. Without a detailed write up on this (a significant revision), I as reviewer cannot change my assessment simply based on a single paragraph here. My original point stands that the evaluation was not biologically meaningful.

  The authors did not update the draft, but contended that the discussion was sufficient.
* Diversity of generated sequences particularly in the qualitative studies: The authors have updated the draft with an additional figure in the pdf demonstrating sequence similarity of RNA sequences generated by LiRDSC. This suggests that the sequences are more diverse than initially thought by reviewers. However, I was unable to find sufficient detail about this experiment. Specifically, it is unclear to me how exactly these are generated. e.g. is this unconditional, conditional, or a combination? In my mind given other experiments the extent to which this model exhibits mode collapse, overfitting, and sequence diversity issues is still unclear. Given the size of the dataset, this is one of the major concerns in this area.

* Benefits of diffusion noising: Reviewer 2YkC inquired about the diffusion noising and its effectiveness. The authors showed ablations on the number of steps and promised further experiments in the updated draft. I was not able to find these experiments leaving this as an open question.

Other concerns seem sufficiently addressed in the rebuttal.

**Reviewer Scores:**

EeHz: No change

gNL1: Unlikely to change given overall positive opinion and lingering concerns around novelty which are true, but not grounds for rejection on their own

2YkC: Possible, but unlikely to change their score without the ablation they requested.

XHfM: This reviewer responded
> If the authors could convincingly show that the model is capable of sampling multiple diverse sequences per target that are close to solving the tasks in 3D space (while addressing the obvious flaws in the manuscript to make it appear more professional), I would lean towards acceptance.

I don't believe this bar has been met with the current experiments, and therefore believe the reviewer would not have changed their score without further clarifications.However, it is possible given further discussion that this reviewer would have changed to a score of 6.

Given this analysis I recommend rejection of this paper at this time. I believe this paper, including its dataset and model, could be quite useful to the ML community, however concerns remain that need to be clarified in an updated version of the paper.

---

### Decision · Program_Chairs · 2026-01-26

Reject